# Natural Counterfactuals With Necessary Backtracking

**Guang-Yuan Hao**[1,5]*, **Jiji Zhang**[1]*, **Biwei Huang**[2], **Hao Wang**[3], **Kun Zhang**[4,5]

[1]Chinese University of Hong Kong, [2]University of California San Diego, [3]Rutgers University
[4]Carnegie Mellon University, [5]Mohamed bin Zayed University of Artificial Intelligence
`guangyuanhao@outlook.com, jijizhang@cuhk.edu.hk`
`bih007@ucsd.edu, hw488@cs.rutgers.edu, kunz1@cmu.edu`

## Abstract

Counterfactual reasoning is pivotal in human cognition and especially important for providing explanations and making decisions. While Judea Pearl's influential approach is theoretically elegant, its generation of a counterfactual scenario often requires too much deviation from the observed scenarios to be feasible, as we show using simple examples. To mitigate this difficulty, we propose a framework of *natural counterfactuals* and a method for generating counterfactuals that are more feasible with respect to the actual data distribution. Our methodology incorporates a certain amount of backtracking when needed, allowing changes in causally preceding variables to minimize deviations from realistic scenarios. Specifically, we introduce a novel optimization framework that permits but also controls the extent of backtracking with a "naturalness" criterion. Empirical experiments demonstrate the effectiveness of our method. The code is available at `https://github.com/GuangyuanHao/natural_counterfactuals`.

## 1 Introduction

Counterfactual reasoning, which aims to answer what a feature of the world would have been if some other features had been different, is often used in human cognition, to perform self-reflection, provide explanations, and inform decisions [29, 6]. For AI systems to achieve human-like abilities of reflection and decision-making, incorporating counterfactual reasoning is crucial. Judea Pearl's structural approach to counterfactual modeling and reasoning [26] has been especially influential in recent decades. Within this framework, counterfactuals are conceptualized as being generated by surgical interventions on the variables to be changed, while leaving its causally upstream variables intact and all downstream causal mechanisms invariant. These counterfactuals are thoroughly non-backtracking in that the desired change is supposed to happen without tracing back to changes in causally preceding variables. Reasoning about these counterfactuals can yield valuable insights into the consequences of hypothetical actions. Consider a scenario: a sudden brake of a high-speed bus caused Tom to fall and injure Jerry. Non-backtracking counterfactuals would tell us that if Tom had stood still (despite the sudden braking), then Jerry would not have been injured. Pearl's approach supplies a principled machinery to reason about conditionals of this sort, which are usually useful for explanation, planning, and responsibility allocation.

However, such surgical interventions are sometimes so removed from what is or can be observed that it is difficult or even impossible to learn from data the consequences of such interventions. In the previous example, preventing Tom's fall in a sudden braking scenario requires defying mechanisms that are difficult or even physically impossible to disrupt. As a result, there are likely to be no data points in the reservoir of observed scenarios that are consistent with a person standing still during a

---

*The first two authors contributed equally to this work.

sudden braking. If so, it can be very challenging to learn to generate such a counterfactual from the available data, as we will demonstrate in our experiments.[2]

In this paper, we introduce a notion of "natural counterfactuals" to address the above issue with non-backtracking counterfactuals. Our notion will allow a certain amount of backtracking, to keep the counterfactual scenario "natural" with respect to the available observations. For example, rather than the unrealistic scenario where Tom does not fall at a sudden bus stop, a more natural counterfactual scenario to realize the change to not-falling would involve changing at the same time some causally preceding events, such as changing the sudden braking to gradually slowing down. On the other hand, our notion also constrains the extent of backtracking; in addition to a naturalness criterion, we formulate an optimization scheme to encourage minimizing backtracking while meeting the naturalness criterion.

As we will show empirically, this new notion of counterfactual is especially useful from a machine learning perspective. When interventions lead to unrealistic scenarios relative to the training data, predicting counterfactual outcomes in such scenarios can be highly uncertain and inaccurate [13]. This issue becomes particularly pronounced when non-parametric models are employed, as they often struggle to generalize to unseen, out-of-distribution data [31]. The risk of relying on such counterfactuals is thus substantial, especially in high-stake applications like healthcare and autonomous driving. In contrast, our approach amounts to searching for feasible changes that keep generated counterfactuals within their original distribution, employing backtracking when needed. This strategy effectively reduces the risk of inaccurate predictions and ensures more reliable results.

In short, our approach aims to achieve the goal of ensuring that counterfactual scenarios remain sufficiently realistic with respect to the actual data distribution by permitting minimal yet necessary backtracking. It is designed with two major elements. First, we need criteria to determine the feasibility of interventions, ensuring they are realistic with respect to the actual data distribution. Second, we appeal to backtracking when and only when it is necessary to avoid infeasible interventions, and need to develop an optimization framework to realize this strategy. To be clear, our aim is not to propose uniquely correct semantics for counterfactuals or to show that our notion of counterfactuals is superior to others for all purposes. Rather, our goal is to develop a notion of counterfactual that is theoretically well-motivated and practically useful in some contexts and for certain purposes. There may in the end be a general semantics to unify fruitfully our notion and other notions of counterfactual in the literature, but we will not attempt that in this paper. Our working assumption is that there can be different but equally coherent notions of counterfactuals, and we aim to show that the notion developed in this paper is particularly useful for generating counterfactual instances that are feasible with respect to data.

The key contributions of this paper include:

- Developing a more flexible and realistic notion of natural counterfactuals, addressing the limitations of non-backtracking reasoning while keeping its merits as far as possible.
- Introducing an innovative and feasible optimization framework to generate natural counterfactuals.
- Detailing a machine learning approach to produce counterfactuals within this framework, with empirical results from simulated and real data showcasing the superiority of our method compared to non-backtracking counterfactuals.

## 2 Related Work

**Non-backtracking Counterfactual Generation.** As will become clear, our theory is presented in the form of counterfactual sampling or generation. [28, 18, 8, 30] use the deep generative models to learn a causal model from data given a causal graph, and then use the model to generate non-backtracking counterfactuals. Our experiments will examine some of these models and demonstrate their difficulties in dealing with interventions that are unrealistic relative to training data, due to the fact that the causal model learned from data is not reliable in handling inputs that are out-of-distribution.

---

[2]Moreover, one may argue that the envisaged counterfactual may be too unnatural to be relevant for practical purposes. For example, from a legal perspective, Tom's causing Jerry's injury could be given a "necessity defense," acknowledging that the sudden braking left him with no alternatives [7]. Hence, for the purpose of allocating responsibility, reasoning about the counterfactual situation of Tom standing still despite the sudden braking is perhaps irrelevant or even misleading.

**Backtracking Counterfactuals.** Backtracking in counterfactual reasoning has drawn plenty of attention in philosophy [14], psychology [9], and cognitive science [12]. [14] proposes a theory that is in a spirit similar to ours, in which backtracking is allowed but limited by some requirement of matching as much causal upstream as possible. [12] shows that people use both backtracking and non-backtracking counterfactuals in practice and tend to use backtracking counterfactuals when explicitly required to explain causes for the supposed change in a counterfactual. [35] is the first work, as far as we know, to formally introduce backtracking counterfactuals in the framework of structural causal models. The main differences between this work and ours are that [35] requires "full" backtracking given a causal model, all the way back to exogenous noise terms, and measures closeness in terms of these noise terms, whereas we limit backtracking to what is needed to render the counterfactual "natural", and the measure of closeness in our framework can be defined directly on endogenous, observed variables, which is arguably desirable because changes to the unobserved variables are by definition outside of our control and not actionable. More detailed comparisons can be found in Sec. F of the Appendix.

**Algorithmic Recourse and Counterfactual Explanations.** Algorithmic recourse (AR) [16] and counterfactual explanations (CE) [36, 10, 23, 3, 24, 34, 32, 37] represent two leading strategies within explainable AI that heavily exploit counterfactual reasoning or reasoning about intervention effects. Various studies [16, 1, 27, 15, 37] explore concepts of feasibility akin to that of "naturalness" in this work, though apply them in quite different contexts. The objectives of CE and AR are to pinpoint minimal alterations to an input (in CE) or minimal interventions (in AR) that would either induce a desirable output from a predictive model (in CE) or lead to a desirable outcome (in AR). In contrast, our work here focuses primarily on generating counterfactuals. While this paper does not directly address CE or AR, our notion of natural counterfactuals is likely to be very relevant to these tasks.

## 3 Preliminaries

In this section, we begin by outlining various basic concepts in causal inference, followed by an introduction to non-backtracking counterfactuals.

**Structural Causal Models (SCM).** We assume there is an underlying recursive SCM [26] of the following sort to represent the data generating process. An SCM $\mathcal{M} := \langle \mathbf{U}, \mathbf{V}, \mathbf{f}, p(\mathbf{U}) \rangle$ consists of exogenous (noise) variables $\mathbf{U} = \{\mathbf{U}_1, ..., \mathbf{U}_N\}$, endogenous (observed) variables $\mathbf{V} = \{\mathbf{V}_1, ..., \mathbf{V}_N\}$, functions $\mathbf{f} = \{f_1, ..., f_N\}$, and a joint distribution $p(\mathbf{U})$ of noise variables, which are assumed to be jointly independent. Each function, $f_i \in \mathbf{f}$, specifies how an endogenous variable $\mathbf{V}_i$ is determined by its parents $\mathbf{PA}_i \subseteq \mathbf{V}$:

$$\mathbf{V}_i := f_i(\mathbf{PA}_i, \mathbf{U}_i), \quad i = 1, ..., N \tag{1}$$

Such an SCM entails a (causal) Bayesian network over the observed variables, consisting of the directed acyclic graph over $\mathbf{V}$ in which there is an arrow from each member of $\mathbf{PA}_i$ to $\mathbf{V}_i$, and the joint distribution of $\mathbf{V}$ induced by $\mathbf{f}$ and $p(\mathbf{U})$. **In our setting, we assume this causal graph is known and samples from this joint distribution are available, but $\mathbf{f}$ and $p(\mathbf{U})$ are not given**, though some assumptions on $\mathbf{f}$ and $p(\mathbf{U})$ will be needed for identifiability.

**Local Mechanisms.** Functions in $\mathbf{f}$ are usually regarded as representing local mechanisms. In this paper, however, we will use the term "local mechanism" to refer to the conditional distribution of an endogenous variable given its parent variables, i.e., $p(\mathbf{V}_i | \mathbf{PA}_i)$ for $i = 1, ..., N$, which can be estimated from the available information. Note that a local mechanism in this sense implicitly encodes the properties of the corresponding noise variable; given a fixed value of $\mathbf{PA}_i$, the noise $\mathbf{U}_i$ determines the probability distribution of $\mathbf{V}_i$ [26]. Hence, the term "local mechanism" will also be used sometimes to refer to the distribution of the noise variable $p(\mathbf{U}_i)$.

**Intervention**. Given an SCM, an intervention on a set of endogenous variables $\mathbf{A} \subseteq \mathbf{V}$ is represented by replacing the functions for members of $\mathbf{A}$ with constant functions $X = x^*$, where $X \in \mathbf{A}$ and $x^*$ is the target value of $X$, and leaving the functions for other variables intact [26].[3]

**Non-Backtracking Counterfactuals.** Let $\mathbf{A}$, $\mathbf{B}$, and $\mathbf{E}$ be sets of endogenous variables. A general counterfactual question takes the following form: given evidence $\mathbf{E} = \mathbf{e}$, what would the value of $\mathbf{B}$ have been if $\mathbf{A}$ had taken the value setting $\mathbf{a}^*$? In this paper, we focus on a special case of this

---

[3]Following a standard notation, we use a $*$ superscript to signal counterfactual values.

question in which $\mathbf{E} = \mathbf{V}$, i.e., the evidence is a complete data point covering all observed variables. That is, given an actual data point, we consider what the data point would have been if a variable had taken a different value than its actual value. The Pearlian, non-backtracking reading of this question takes the counterfactual supposition of $\mathbf{A} = \mathbf{a}^*$ to be realized by an intervention on $\mathbf{A}$ [26]. This means that in the envisaged counterfactual scenario, all variables in the causal upstream of $\mathbf{A}$ keep their actual values while $\mathbf{A}$ takes a different value. As mentioned, a potential problem is that such a scenario is outside of the support of available data, and so it is often unreliable to make inferences about the downstream variables in the scenario based on the available data.

## 4 A Framework for Natural Counterfactuals

$Do(\cdot)$ **and** $Change(\cdot)$ **Operators.** Using Pearl's influential do-operator, the non-backtracking mode appeals to $do(\mathbf{A} = \mathbf{a}^*)$, an intervention to set the value $\mathbf{A}$ to $\mathbf{a}^*$, to generate a counterfactual instance. However, in our framework, the counterfactual supposition is not necessarily realized by an intervention on $\mathbf{A}$, while keeping all its causal upstream intact. Instead, when $do(\mathbf{A} = \mathbf{a}^*)$ results in a counterfactual setting that violates a naturalness criterion, some backtracking will be invoked. To differentiate from the intervention $do(\mathbf{A} = \mathbf{a}^*)$, we will often use $change(\cdot)$ and write $change(\mathbf{A} = \mathbf{a}^*)$ to denote a desired modification in $\mathbf{A}$. Different semantics for counterfactuals correspond to different interpretations of the change operator. In this paper, we explore an interpretation that connects the change operator to the do operator in a relatively straightforward manner.

The basic idea is that $change(\mathbf{A} = \mathbf{a}^*)$ will correspond to $do(\mathbf{C} = \mathbf{c}^*)$ for some set $\mathbf{C}$ that includes $\mathbf{A}$ and possibly some of $\mathbf{A}$'s causal ancestors. When $\mathbf{C} = \{\mathbf{A}\}$, this is equivalent to a non-backtracking interpretation. In general, however, some variables in $\mathbf{A}$'s causal upstream need to change together with $\mathbf{A}$ in order to keep the counterfactual scenario within the relevant support. A central component of our approach is to design a way to determine $\mathbf{C}$ and $\mathbf{c}^*$, given a request of $change(\mathbf{A} = \mathbf{a}^*)$. We will call the resulting $do(\mathbf{C} = \mathbf{c}^*)$ the **least-backtracking feasible (LBF) intervention** for $change(\mathbf{A} = \mathbf{a}^*)$. Once the LBF intervention is determined, inferences can be made in the same fashion as in Pearl's approach [26].

To determine the LBF intervention for $change(\mathbf{A} = \mathbf{a}^*)$, we formulate it as an optimization problem to search for a minimal change of $\mathbf{A}$'s causal ancestors that, together with changing $\mathbf{A}$ to $\mathbf{a}^*$, satisfy a "naturalness" criterion. Let $\mathbf{AN}(\mathbf{A})$ denote the set of $\mathbf{A}$'s ancestors in the given causal graph together with $\mathbf{A}$ itself. We define the optimization framework, **F**easible **I**ntervention **O**ptimization (FIO), as follows:

$$
\begin{aligned}
\underset{an(\mathbf{A})^*}{\text{minimize}} \quad & D(an(\mathbf{A}), an(\mathbf{A})^*) \\
\text{s.t.} \quad & \mathbf{A} = \mathbf{a}^*, \\
& g_n(an(\mathbf{A})^*) > \epsilon.
\end{aligned}
\tag{2}
$$

where $an(\mathbf{A})$ and $an(\mathbf{A})^*$ represent the actual value setting and the counterfactual value setting of $\mathbf{AN}(\mathbf{A})$ respectively (note that $\mathbf{A} \subseteq \mathbf{AN}(\mathbf{A})$). $g_n(\cdot)$ measures the naturalness of the counterfactual value setting of $\mathbf{AN}(\mathbf{A})$ and $\epsilon$ is a small constant. So the optimization has a naturalness criterion as a constraint. On the other hand, $D(\cdot)$ is a distance metric designed to encourage the counterfactual value setting to invoke the least amount of backtracking. Below we develop these two components in Sec. 4.1 and Sec. 4.2, respectively. Once we obtain the counterfactual value $an(\mathbf{A})^*$ by FIO, the variable set for the LBF intervention includes $\mathbf{A}$ and other variables corresponding to the difference between $an(\mathbf{A})^*$ and $an(\mathbf{A})$, i.e., $\mathbf{A}$ is always included in $\mathbf{C}$ even when $an(\mathbf{A})^* = an(\mathbf{A})$.

### 4.1 Naturalness Constraints

As indicated previously, the intended purpose of the naturalness constraint is to confine the counterfactual instance sufficiently within the support of the data distribution. Roughly and intuitively, the more frequently a value occurs, the more it is considered to be "natural". Moreover, we would like to use a measure of naturalness that takes into account local mechanisms according to the given causal graph rather than just the joint distribution. Therefore, we propose to assess naturalness by examining the distribution characteristics, such as density, of each variable's value $\mathbf{V}_j = \mathbf{v}_j^*$ given the variable's local mechanism $p(\mathbf{V}_j | pa_j^*)$, where $\mathbf{V}_j \in \mathbf{AN}(\mathbf{A})$ and $pa_j^*$ denotes its parental value setting.[4]

---

[4]The notion of "naturalness" can have various interpretations. In our context, it is defined by the observed data distribution, as we assume we can only access observed data and intend the counterfactuals to be empirically

### 4.1.1 Local Naturalness Criteria

We start by proposing some measures of the naturalness of one variable's value, $\mathbf{v}_j^*$, within the counterfactual data point $an(\mathbf{A})^*$ in this section, followed by defining a measure of the overall naturalness of $an(\mathbf{A})^*$ in the next.

Informally, a value satisfies the criterion of *local $\epsilon$-natural generation* if it is a natural outcome of its local mechanism. The proposed measures of naturalness will depend on the specific value $\mathbf{V}_j = \mathbf{v}_j^*$, alongside its parent value $\mathbf{PA}_j = pa_j^*$, noise value $\mathbf{U}_j = \mathbf{u}_j^*$, and the corresponding local mechanism, expressed by $p(\mathbf{V}_j|\mathbf{PA}_j = pa_j^*)$ or $p(\mathbf{U}_j)$. The cumulative distribution function (CDF) for noise variable $\mathbf{U}_j$ at $\mathbf{U}_j = \mathbf{u}_j^*$ is $F(\mathbf{u}_j^*) = \int_{-\infty}^{\mathbf{u}_j^*} p(\mathbf{U}_j)d\mathbf{U}_j$, and for the conditional distribution $p(\mathbf{V}_j|\mathbf{PA}_j = pa_j^*)$ at $\mathbf{V}_j = \mathbf{v}_j^*$ is $F(\mathbf{V}_j|pa_j^*) = \int_{-\infty}^{\mathbf{v}_j^*} p(\mathbf{V}_j = \mathbf{v}_j^*|pa_j^*)d\mathbf{V}_j$.

We propose the following potential criteria based on entropy-normalized density, CDF of exogenous variables, and CDF of conditional distributions, respectively. The entropy-normalized naturalness measure evaluates the naturalness of $\mathbf{v}_j^*$ in relation to its local mechanism $p(\mathbf{V}_j|\mathbf{PA}_j = pa_j^*)$. The CDF-based measures, namely the latter two criteria, consider data points in the tails to be less natural. Each of these criteria has its own intuitive appeal, and their relative merits will be discussed subsequently. Below, we use $g_l(\mathbf{v}_j^*)$ to represent a (local) naturalness measure of $\mathbf{v}_j^*$. We consider the following three possible measures:

(1) Entropy-Normalized Measure: $g_l(\mathbf{v}_j^*) = p(\mathbf{v}_j^*|pa_j^*)e^{H(\mathbf{V}_j|pa_j^*)}$, where $H(\mathbf{V}_j|pa_j^*) = \mathbb{E}[-\log p(\mathbf{V}_j|pa_j^*)]$;

(2) Exogenous CDF Measure: $g_l(\mathbf{v}_j^*) = \min(F(\mathbf{u}_j^*), 1 - F(\mathbf{u}_j^*))$;

(3) Conditional CDF Measure: $g_l(\mathbf{v}_j^*) = \min(F(\mathbf{v}_j^*|pa_j^*), 1 - F(\mathbf{v}_j^*|pa_j^*))$;

where the function $\min(\cdot)$ returns the minimum of the given values. When $g_l(\mathbf{v}_j^*) > \epsilon$, we say the $\mathbf{v}_j^*$ given its causal parents' values satisfies the criterion of local $\epsilon$-natural generation.

Some comments on these choices are in order:

**Choice (1): Entropy-Normalized Measure.** Specifically, Choice (1), $p(\mathbf{v}_j^*|pa_j^*)e^{H(\mathbf{V}_j|pa_j^*)}$, can be rewritten as $e^{\log p(\mathbf{v}_j^*|pa_j^*)+\mathbb{E}[-\log p(\mathbf{V}_j|pa_j^*)]}$, where $-\log p(\mathbf{v}_j^*|pa_j^*)$ can be seen as the measure of surprise of $\mathbf{v}_j^*$ given $pa_j^*$ and $\mathbb{E}[-\log p(\mathbf{V}_j|pa_j^*)]$ can be considered as the expectation of surprise of the local mechanism $p(\mathbf{V}_j|pa_j^*)$ [2]. Hence, the measure quantifies the relative naturalness (i.e., negative surprise) of $\mathbf{V}_j$. Implementing this measure is usually straightforward when employing a parametric SCM where the conditional distributions can be explicitly represented.

**Choice (2): Exogenous CDF Measure.** If using a parametric SCM, we might directly measure differences on exogenous variables. However, in a non-parametric SCM, exogenous variables are in general not identifiable, and different noise variables may have different distributions. Still, we may consider using the CDF of exogenous variables to align the naturalness of different distributions, based on a common assumption for non-parametric SCMs in the machine learning system. The assumption is that the support of the local mechanism $p(\mathbf{V}_j|\mathbf{PA}_j = pa_j^*)$ does not contain disjoint sets, the function $f_j$ in the SCM is monotonically increasing with respect to the noise variable $\mathbf{U}_j$, which is assumed to follow a standard Gaussian distribution [20]. Data points from the tails of a standard Gaussian can be thought of as improbable events. Hence, $\mathbf{V}_j = \mathbf{v}_j^*$ satisfies local $\epsilon$-natural generation when its exogenous CDF $F(\mathbf{u}_j^*)$ falls within the range $(\epsilon, 1 - \epsilon)$. In practice, for a single variable, $\mathbf{U}_j$ is a one-dimensional variable, and it is easier to enforce the measure than Choice (1), which involves conditional distributions.

**Choice (3): Conditional CDF Measure.** The measure treats a particular value in the tails of local mechanism $p(\mathbf{V}_j|pa_j^*)$ as unnatural. Hence, $\mathbf{V}_j = \mathbf{v}_j^*$ meets local $\epsilon$-natural generation when $F(\mathbf{V}_j = \mathbf{v}_j^*|pa_j^*)$ falls within the range $(\epsilon, 1 - \epsilon)$ instead of tails. This measure can be used in parametric models where the conditional distribution can be explicitly represented. It can also be easily used in non-parametric models and the measure is equivalent to Choice (2) when those models

---

supported. While this is obviously not the only plausible interpretation of naturalness, it is a useful one for our purpose.

satisfy the assumption mentioned in Choice (2), since the CDF $F(\mathbf{V}_j|pa_j^*)$ has a one-to-one mapping with the CDF $F(\mathbf{U}_j)$, i.e., $F(\mathbf{v}_j^*|pa_j^*) = F(\mathbf{u}_j^*)$, when $\mathbf{v}_j^* = f(pa_j^*, \mathbf{u}_j^*)$.

#### 4.1.2 Global Naturalness Criteria

Given a local naturalness measure $g_l$, we simply define a global naturalness measure for $an(\mathbf{A})^*$ as

$$g_n(an(\mathbf{A})^*) = \min_{\mathbf{v}_j^* \in an(\mathbf{A})^*} (g_l(\mathbf{v}_j^*)). \tag{3}$$

That is, $g_n(an(\mathbf{A})^*)$ returns the smallest local naturalness value among members of $\mathbf{AN}(\mathbf{A})$. Finally, we can define a criterion of $\epsilon$-natural generation to assess whether the counterfactual value $an(\mathbf{A})^*$ is sufficiently natural.

**Definition 1** ($\epsilon$-**Natural Generation**). *Given an SCM containing a set $\mathbf{A}$, let $\mathbf{AN}(\mathbf{A})$ contain all ancestors of $\mathbf{A}$ and $\mathbf{A}$ itself. A value setting $\mathbf{AN}(\mathbf{A}) = an(\mathbf{A})^*$ satisfies $\epsilon$-natural generation, if and only if, $g_n(an(\mathbf{A})^*) > \epsilon$ and $\epsilon$ is a small constant.*

Obviously, a larger value of $\epsilon$ implies a higher standard for the naturalness of the counterfactual value setting $an(\mathbf{A})^*$. To consider only feasible interventions, we require $an(\mathbf{A})^*$ to meet $\epsilon$-natural generation, which is a constraint used in FIO.

### 4.2 Distance Measure to Limit Backtracking

We now turn to the distance measure in Eqn. 2 of the FIO framework. We considered two distinct distance measures in this work. The first prioritizes minimizing changes in the observed causal ancestors of the target variable of the desired change. The second focuses on reducing alterations in local mechanisms, regarding them as inherent costs of an intervention. Due to space limitations, we will introduce here only the simpler measure in terms of minimal changes in the observable causal ancestors. A discussion of the other measure can be found in Sec. G.

For our purpose, the $L^1$ norm is a good choice, as it encourages sparse changes and thus sparse backtracking:

$$D(an(\mathbf{A}), an(\mathbf{A})^*) = \|an(\mathbf{A})^* - an(\mathbf{A})\|_1 \tag{4}$$

where $an(\mathbf{A})$ and $an(\mathbf{A})^*$ represent the actual value and counterfactual value of $\mathbf{A}$'s ancestors $\mathbf{AN}(\mathbf{A})$ respectively, where $\mathbf{A} \in \mathbf{AN}(\mathbf{A})$. Because endogenous variables may vary in scale, e.g., a normal distribution with a range of $(-\infty, \infty)$ versus a uniform distribution over the interval $[0, 1]$, we standardize each endogenous variable before computing the distance. This normalization ensures a consistent and fair evaluation of changes.

Implicitly, this distance metric favors changes in variables that are proximal to $\mathbf{A}$, since altering a more remote variable typically results in changes to more downstream variables. In the extreme case, when the value $an(\mathbf{A})^*$ corresponding to $do(\mathbf{A} = \mathbf{a}^*)$, i.e., the one corresponding to the non-backtracking counterfactual, already meets the $\epsilon$-natural generation criterion, the distance metric $D(an(\mathbf{A}), an(\mathbf{A})^*)$ will achieve the minimal value $|\mathbf{a} - \mathbf{a}^*|$. In such a case, no backtracking is needed and the non-backtracking counterfactual will be generated. However, If $do(\mathbf{A} = \mathbf{a}^*)$ does not meet the $\epsilon$-natural generation criterion, it becomes necessary to backtrack.

### 4.3 Identifiability of Natural Counterfactuals

As said, we assume we do not have prior knowledge of the functions of the SCM and so noise variables are in general not identifiable from the observed variables. However, if we assume the SCM satisfies the conditions of the following theorem, then the counterfactual instance resulting from an LBF intervention is identifiable.

**Theorem 4.1** (Identifiable Natural Counterfactuals). *Given the causal graph and the joint distribution over $\mathbf{V}$, suppose $\mathbf{V}_i$ satisfies the following structural causal model: $\mathbf{V}_i := f_i(\mathbf{PA}_i, \mathbf{U}_i)$ for any $\mathbf{V}_i \in \mathbf{V}$, assume every $f_i$, though unknown, is smooth and strictly monotonic w.r.t. $\mathbf{U}_i$ for fixed values of $\mathbf{PA}_i$. Then, given an actual data point $\mathbf{V} = \mathbf{v}$, with an LBF intervention $do(\mathbf{C} = \mathbf{c}^*)$ (satisfying the criterion of $\epsilon$-natural generation), the counterfactual instance $\mathbf{V} = \mathbf{v}^*$ is identifiable: $\mathbf{V} = \mathbf{v}^*|do(\mathbf{C} = \mathbf{c}^*), \mathbf{V} = \mathbf{v}$.*

This theorem confirms the identifiability of our natural counterfactuals from the causal graph and the joint distribution over the observed variables.[5] Specifically, since $do(\mathbf{C} = \mathbf{c}^*)$ satisfies the criterion of $\epsilon$-natural generation, it guarantees that the resulting counterfactual instance falls within the support of the observed joint distribution. Then, building on Theorem 1 from [20], we can demonstrate that using the actual data distribution allows for the inference of natural counterfactuals without knowing the functions or the noise distributions of the SCM.

## 5  A Practical Method for Generating Natural Counterfactuals

In this section, we provide a practical method for solving (approximately) the FIO problem described in the last section. We assume that we are given data sampled from the joint distribution of the endogenous variables and a causal graph, and that the underlying SCM satisfies the assumptions in Theorem 4.1. We learn a generative model for the endogenous variables from data, serving as an estimated SCM to generate natural counterfactuals: $\mathbf{V}_i := \hat{f}_i(\mathbf{PA}_i, \mathbf{U}_i)$ for $i = 1, ..., N$, where $\hat{f}_i$ is assumed to be reversible given $\mathbf{PA}_i$. Note that, unlike the functions in the true SCM, these learned functions in general do not generalize well to out-of-distribution data, as demonstrated in the experiments, which, recall, is a main motivation for employing natural counterfactuals instead. For simplicity, we assume the noise distribution is standard Gaussian, though the identifiability of natural counterfactuals only requires that noise variables are continuous and do not depend on the specific form of the noise distribution. The specific FIO problem we target plugs the distance measure (Eqn. 4) and the naturalness measure from Choice (3) in Sec. 4.1.1 (or Choice (2), which is equivalent to Choice (3) given the assumptions) into Eqn. 2:

$$
\begin{aligned}
&\underset{an(\mathbf{A})^*}{\text{minimize}} \; \|an(\mathbf{A})^* - an(\mathbf{A})\|_1 \\
&s.t. \quad \mathbf{A} = \mathbf{a}^*, \\
&\qquad \epsilon < F(\mathbf{V}_j = \mathbf{v}_j^* | pa_j^*) < 1 - \epsilon, \forall \mathbf{V}_j \in \mathbf{AN}(\mathbf{A}).
\end{aligned}
\tag{5}
$$

Again, in theory, there may be no solution for $an(A)^*$ if the naturalness criterion is demanding. In the extreme case, for example, even when $\mathbf{a}^* = \mathbf{a}$, if $\epsilon$ is set so high that the actual instance does not satisfy the condition of $\epsilon$-natural generation, then no solution exists.

We propose to solve this optimization problem using the following approximate method. Since the estimated functions $\hat{f}_j$ are assumed to be reversible, we can reformulate the problem of searching for optimal values of the endogenous variables as one of searching for optimal values of the exogenous noise variables. A feasible approach is to use the Lagrangian method [5] to minimize the following objective loss:

$$
\mathcal{L}(\mathbf{u}_{\mathbf{AN}(\mathbf{A})}^*) = \sum_{\mathbf{u}_j^* \in \mathbf{u}_{\mathbf{AN}(\mathbf{A})}^*} |\hat{f}_j(pa_j^*, \mathbf{u}_j^*) - \mathbf{v}_j| + w_\epsilon \sum_{\mathbf{u}_j^* \in \mathbf{u}_{\mathbf{AN}(\mathbf{A})}^*} [\max(\epsilon - F(\mathbf{u}_j^*), 0) + \max(\epsilon + F(\mathbf{u}_j^*) - 1, 0)]
$$
$$
s.t. \quad \mathbf{u}_{\mathbf{A}}^* = \hat{f}_{\mathbf{A}}^{-1}(pa_{\mathbf{A}}^*, \mathbf{a}^*)
\tag{6}
$$

where the optimization parameters are the counterfactual values of the noise variables corresponding to $\mathbf{A}$'s ancestors, $\mathbf{u}_{\mathbf{AN}(\mathbf{A})}^*$, and the function $\max(\cdot)$ returns the maximum between two values. The first term is the distance measure in the FIO problem, while the second term implements the constraint of $\epsilon$-natural generation. The hyperparameter $w_\epsilon$ serves to modulate the penalty imposed on non-natural values. Notice that, in order to ensure the hard constraint $\mathbf{A} = \mathbf{a}^*$, $\mathbf{A}$'s noise value $\mathbf{u}_{\mathbf{A}}^*$ is not optimized explicitly, since the value $pa_{\mathbf{A}}^*$ is fully determined by $\mathbf{a}^*$ and other noise values. Hence, only noise values other than $\mathbf{u}_{\mathbf{A}}^*$ are optimized. Further details are provided in Sec. D.

For simplicity, we have focused on the case of "full evidence" in presenting our framework and method. But it is straightforward to extend the approach to address cases of "partial evidence", i.e., $\mathbf{E} \neq \mathbf{V}$. In such cases, we can simply sample from $p(\mathbf{V}|\mathbf{E})$, treating it as full evidence, and obtain a natural counterfactual from the resulting sample.

---

[5]Given a substantial naturalness constraint, there may be one or more solutions for $\mathbf{C}$, or no solution at all. Which of these is the case can always be identified. The above theorem on identifiability of natural counterfactuals is concerned only with the cases in which a natural counterfactual exists, meaning that there is at least one solution for $\mathbf{C}$. The proof of Theorem 4.1 is provided in Sec. B.

Table 1: MAE Results on *Toy 1* to *Toy 4* (Lower MAE is better). To save room, we also write "*do*" for "*change*" for natural counterfactuals.

| Dataset | Toy 1 | | | Toy 2 | Toy 3 | | | | | | Toy 4 | | |
|---|---|---|---|---|---|---|---|---|---|---|---|---|---|
| $do$ or $change$ | $do(n_1)$ | | $do(n_2)$ | $do(n_1)$ | $do(n_1)$ | | | $do(n_2)$ | | $do(n_3)$ | $do(n_1)$ | | $do(n_2)$ |
| Outcome | $n_2$ | $n_3$ | $n_3$ | $n_2$ | $n_2$ | $n_3$ | $n_4$ | $n_3$ | $n_4$ | $n_4$ | $n_2$ | $n_3$ | $n_3$ |
| Non-backtracking | 0.477 | 0.382 | 0.297 | 0.315 | 0.488 | 0.472 | 0.436 | 0.488 | 0.230 | 0.179 | 0.166 | 0.446 | 0.429 |
| Ours | 0.434 | 0.354 | 0.114 | 0.303 | 0.443 | 0.451 | 0.423 | 0.127 | 0.136 | 0.137 | 0.158 | 0.443 | 0.327 |

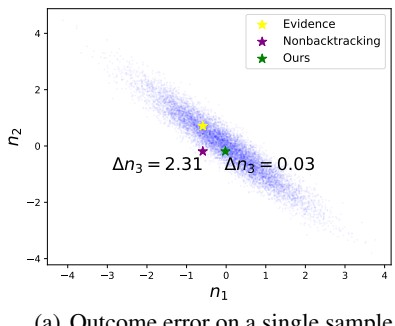

(a) Outcome error on a single sample

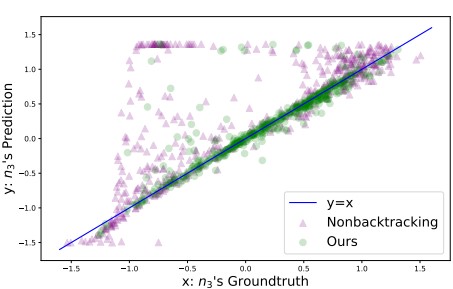

(b) Groundtruth-Prediction Scatter Plot

Figure 1: The Visualization Results on *Toy 1* (View the enlarged figure in Fig. 3 in the Appendix).

# 6 Experiments

In this section, we evaluate the effectiveness of our method through empirical experiments on four synthetic datasets and two real-world datasets.

We propose using the deviations between generated and ground-truth outcomes as a measure of performance. We expect our natural counterfactuals to significantly reduce errors compared to non-backtracking counterfactuals. This advantage can be attributed to the effectiveness of our method in performing necessary backtracking that identifies feasible interventions, keeping counterfactual values within the data distribution, so that the learned functions are applicable. On the other hand, non-backtracking counterfactuals often produce out-of-distribution values [13, 31], posing challenges for generalization using the learned functions.

## 6.1 Simulation Experiments

We start with four simulation datasets, which we use designed SCMs to generate. Please refer to the Appendix for more details about these datasets. Let's first look at *Toy 1*, which contains three variables $(n_1, n_2, n_3)$. $n_1$ is the confounder of $n_2$ and $n_3$, and $n_1$ and $n_2$ cause $n_3$.

**Experimental Settings.** Again, we assume data and a causal graph are known, but not the ground-truth SCM. We employ normalizing flows to learn a generative model of variables $(n_1, n_2, n_3)$ compatible with the causal order. Given the pretrained causal model and a data point from the test set as evidence, we set $n_1$ or $n_2$ as **A** and randomly sample values from test dataset as counterfactual values of the target variable $n_1$ or $n_2$. In our natural counterfactuals, we use Eqn. 6 to determine LBF interventions, with $\epsilon = 10^{-4}$ and $w_\epsilon = 10^4$, while in non-backtracking counterfactuals, $n_1$ or $n_2$ is directly intervened on. We report the Mean Absolute Error (MAE) between our learned counterfactual outcomes and ground-truth outcomes on $n_2$ or/and $n_3$ with multiple random seeds. Notice there may be no feasible interventions for some changes, as we mentioned in Sec. 5, and thus we only report outcomes with feasible interventions, which are within the scope of our natural counterfactuals.

**Visualization of Counterfactuals on a Single Sample.** We assess the counterfactual outcomes for a sample $(n_1, n_2, n_3) = (-0.59, 0.71, -0.37)$, given the desired alteration $change(n_2 = 0.19)$. For natural counterfactuals, it is necessary to backtrack to $n_1$ to realize the change. This step ensures

that the pair $(n_1, n_2)$ remains within a high-density area of the data distribution. In essence, our intervention targets the composite variable $C = (n_1, n_2)$. On the other hand, for non-backtracking counterfactual, the intervention simply modifies $n_2$ to 0.19 without adjusting $n_1$, making $(n_1, n_2)$ out of data support. In Fig. 1 (a), we depict the original data point (yellow), the non-backtracking counterfactual (purple), and the natural counterfactual (green) for $(n_1, n_2)$. The ground-truth support for these variables is shown as a blue scatter plot.

*(1) Feasible Intervention VS Hard Intervention.* Non-backtracking counterfactuals apply a hard intervention on $n_2$ ($do(n_2 = 0.19)$), shifting the evidence (yellow) to the post-intervention point (purple), which lies outside the support of $(n_1, n_2)$. This shows that direct interventions can result in unnatural values. Conversely, our natural counterfactual (green) remains within the support of $(n_1, n_2)$ due to backtracking and the LBF intervention on $(n_1, n_2)$.

*(2) Outcome Error.* We calculate the absolute error between $n_3$'s model prediction and ground-truth value using either the green or purple point as input for the model $p(n_3|n_1, n_2)$. The error for the green point is significantly lower at $0.03$, compared to $2.31$ for the purple point. This lower error with the green point is because it stays within the data distribution after an LBF intervention, allowing for better model generalization than the out-of-distribution purple point.

**Counterfactuals on Whole Test Set.** In Fig. 1 (b), we illustrate the superior performance of our counterfactual method on the test set, notably outperforming non-backtracking counterfactuals. This is evident as many outcomes from non-backtracking counterfactuals for $n_3$ significantly diverge from the $y = x$ line, showing a mismatch between predicted and ground-truth values. In contrast, our method's outcomes largely align with this line, barring a few exceptions possibly due to learned model's imperfections. This alignment is attributed to our method's consistent and feasible interventions, enhancing prediction accuracy, while non-backtracking counterfactuals often lead to infeasible results. Table 1 supports these findings, demonstrating that our approach exhibits an MAE reduction of $61.6\%$ when applied to $n_2$, compared with the non-backtracking method. *Furthermore, our method excels even when intervening in the case of $n_1$, a root cause, by excluding points that do not meet the $\epsilon$-natural generation criteria, further demonstrating its effectiveness.*

**Additional Causal Graph Structures.** Our method also shows superior performance on three other simulated datasets with varied causal graph structures (*Toy 2* to *Toy 4*), as demonstrated in Table 1.

## 6.2 MorphoMNIST

As depicted in Fig. 4 (a) of Appendix, $t$ (digit stroke thickness) causes both $i$ (stroke intensity) and $x$ (images), with $i$ being the direct cause of $x$ in MorphoMNIST. In this experiment, mirroring those in Section 6.1, we incorporate two key changes. First, we utilize two advanced deep learning models, V-SCM [25] and H-SCM [28]. Although both models are referred to as "SCM," they only learn the conditional distributions of endogenous variables and, in theory, do not capture the functional relationships for out-of-distribution inputs. Second, due to the absence of ground-truth SCM for assessing outcome error, we adopt the **counterfactual effectiveness** metric from [28, 22]. This involves training a predictor on the dataset to estimate parent values $(\hat{t}, \hat{i})$ from a counterfactual image $x$ generated by model $p(x|t, i)$ with the input $(t, i)$, and then computing the absolute error $|t - \hat{t}|$ or $|i - \hat{i}|$.

Table 2: Ablation Study on $\epsilon$ (Lower MAE is better)

| Model | $\epsilon$ | CFs | do(t) | | do(i) | |
|---|---|---|---|---|---|---|
| | | | $t$ | $i$ | $t$ | $i$ |
| V-SCM | - | NB | 0.336 | 4.532 | 0.283 | 6.556 |
| | $10^{-4}$ | | 0.314 | 4.506 | 0.171 | 4.424 |
| | $10^{-3}$ | Ours | 0.298 | 4.486 | 0.161 | 4.121 |
| | $10^{-2}$ | | 0.139 | 4.367 | 0.145 | 3.959 |
| H-SCM | - | NB | 0.280 | 2.562 | 0.202 | 3.345 |
| | $10^{-4}$ | | 0.260 | 2.495 | 0.105 | 2.211 |
| | $10^{-3}$ | Ours | 0.245 | 2.442 | 0.096 | 2.091 |
| | $10^{-2}$ | | 0.093 | 2.338 | 0.083 | 2.063 |

**Ablation Study on Naturalness Threshold $\epsilon$.** Table 2 demonstrates that our error decreases with increasing $\epsilon$, regardless of whether V-SCM or H-SCM is used. This trend suggests that a larger $\epsilon$ sets a stricter standard for naturalness in counterfactuals, enhancing the feasibility of interventions and consequently lowering prediction errors. This improvement is due to that deep-learning models are more adept at generalizing to high-frequency data [11].

### 6.3 3DIdentBOX

In this study, we employ two practical public datasets from 3DIdent-BOX [4], namely Weak-3DIdent and Strong-3DIdent, where each image contains a teapot. Both datasets share the same causal graph, as depicted in Fig. 7 (b) of the Appendix, which includes an image variable $x$ and its seven parent variables, with a single variable $b$ and three pairs of variables: $(h, d)$, $(v, \beta)$, and $(\alpha, \gamma)$, where one is the direct cause of the other in each pair.

Table 3: MAE Results on Weak-3DIdent and Strong-3DIdent (abbreviated as "Weak" "Strong" for simplicity). Lower MAE is better. For clarity, we use "Non" to denote the non-backtracking.

| Dataset | - | $d$ | $h$ | $v$ | $\gamma$ | $\alpha$ | $\beta$ | $b$ |
|---------|------|-------|-------|-------|-------|-------|-------|--------|
| Weak | Non | 0.025 | 0.019 | 0.035 | 0.364 | 0.27 | 0.077 | 0.0042 |
| | Ours | 0.024 | 0.018 | 0.034 | 0.349 | 0.221 | 0.036 | 0.0041 |
| Stong | Non | 0.100 | 0.083 | 0.075 | 0.387 | 0.495 | 0.338 | 0.0048 |
| | Ours | 0.058 | 0.047 | 0.050 | 0.298 | 0.316 | 0.139 | 0.0047 |

The primary distinction between Weak-3DIdent and Strong-3DIdent lies in the strength of the causal relationships between each variable pair, with Weak-3DIdent exhibiting weaker connections (Fig. 7 (c)) compared to Strong-3DIdent (Fig. 7 (d)). Our approach mirrors the MorphoMNIST experiments, using H-SCM as the pretrained causal model with $\epsilon = 10^{-3}$.

**Influence of Causal Strength.** As Table 3 reveals, our method outperforms non-backtracking on both datasets, with a notably larger margin in Strong-3DIdent. This increased superiority is due to a higher incidence of infeasible hard interventions in non-backtracking counterfactuals within the Strong-3DIdent dataset.

**See Appendix for More Details.** Please refer to the Appendix for information on datasets, generated samples on MorphoMNIST and 3DIdentBox, standard deviation of results, settings of model training and FIO, differences between our natural counterfactuals and related works, and more.

## 7 Conclusion

Given a non-parametric SCM learned from data and a causal graph, non-backtracking counterfactual inference or generation may be highly unreliable because the corresponding non-backtracking intervention can result in a scenario that is far removed from the data based on which the SCM is learned. To address this issue, we have proposed a notion of natural counterfactuals, which incorporates a naturalness constraint and aims to keep the counterfactual supposition within the support of the training data distribution with minimal backtracking. We also developed a practical method for the generation or inference of natural counterfactuals, the effectiveness of which was demonstrated by empirical results.

In case it is not already transparent, we have not shown, nor did we intend to argue, that natural counterfactuals are superior to non-backtracking counterfactuals for all purposes. Non-backtracking counterfactuals, when correctly inferred, have clear advantages in revealing causal relations between events and the effects of specific actions. However, as we showed in this paper, in the context of data-driven counterfactual reasoning, inference or generation of non-backtracking counterfactuals often go astray due to the challenges of out-of-distribution generalization. Although correct information about non-backtracking counterfactuals has considerable action-guiding values, using unreliable information for that purpose is epistemologically and ethically dubious. We intend our framework of natural counterfactuals to mitigate this kind of risk, though it may appear less elegant theoretically.

Our current method is based on the assumption that the learned functions are invertible. One purpose of using the assumption is to ensure the identifiability of natural counterfactuals when one only has access to endogenous variables. If the assumption does not hold, identifiability is not guaranteed. For example, suppose $Y = XU_1 + U_2$ where $Y$ and $X$ are endogenous variables and $U_1$ and $U_2$ are exogenous noises, then the counterfactual outcome will not be identifiable. However, if we also assume a known distribution of exogenous variables, then our method can be generalized without assuming invertible functions or independent exogenous variables.

Finally, we hasten to reiterate that we do not claim that the particular distance measures and naturalness measures used in this paper are the only choices or among the best. It will be interesting to study and compare alternative implementations in future work.

## Acknowledgment

This material is based upon work supported by the RGC of Hong Kong under GRF13602720, NSF Award No. 2229881, AI Institute for Societal Decision Making (AI-SDM), the National Institutes of Health (NIH) under Contract R01HL159805, and grants from Salesforce, Apple Inc., Quris AI, Florin Court Capital, and the MBZUAI-WIS grant.

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

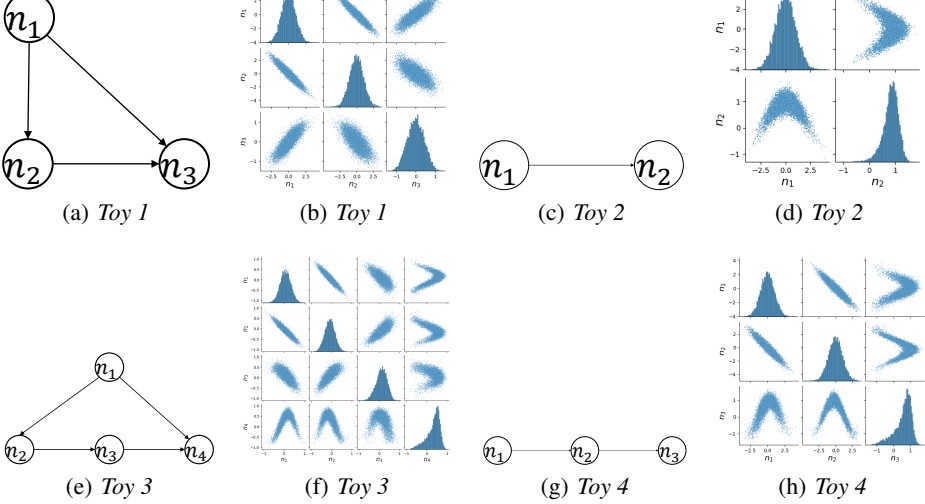

(a) *Toy 1*      (b) *Toy 1*      (c) *Toy 2*      (d) *Toy 2*

(e) *Toy 3*      (f) *Toy 3*      (g) *Toy 4*      (h) *Toy 4*

Figure 2: Causal graphs and Scatter Plot Matrices of *Toy 1-4*. Figure (a) (c) (e) and (g) show causal graphs of *Toy 1-4* respectively. Figures (b) (d) (f) and (h) indicate scatter plot matrices of variables in *Toy 1-4* respectively.

# A  Datasets and More Experimental Results

In this section, we first provide detailed dataset settings and additional experimental results. Subsequently, we present the standard deviation of all experimental outcomes in Sec. A.

## A.1  Toy Datasets

We design four simulation datasets, *Toy 1-4*, and use the designed SCMs to generate $10,000$ data points as a training dataset and another $10,000$ data points as a test set for each dataset. Fig. 2 shows causal graphs of *Toy 1-4* and scatter plot matrices of test datasets in each dataset. The ground-truth SCMs of each dataset are listed below.

*Toy 1*.

$$n_1 = u_1\,, \qquad\qquad\qquad u_1 \sim \mathcal{N}(0,1)\,,$$
$$n_2 = -n_1 + \frac{1}{3}u_2\,, \qquad\qquad u_2 \sim \mathcal{N}(0,1)\,,$$
$$n_3 = \sin\left[0.25\pi(0.5n_2 + n_1)\right] + 0.2u_3\,, \quad u_3 \sim \mathcal{N}(0,1)\,,$$

where there are three endogenous variables $(n_1, n_2, n_3)$ and three noise variables $(u_1, u_2, u_3)$. $n_1$ is the confounder of $n_2$ and $n_3$. $n_1$ and $n_2$ cause $n_3$.

*Toy 2*.

$$n_1 = u_1\,, \qquad\qquad\qquad u_1 \sim \mathcal{N}(0,1)\,,$$
$$n_2 = \sin\left[0.2\pi(n_2 + 2.5)\right] + 0.2u_2\,, \quad u_2 \sim \mathcal{N}(0,1)\,,$$

where there are two endogenous variables $(n_1, n_2)$ and two noise variables $(u_1, u_2)$. $n_1$ causes $n_2$.

*Toy 3*.

$$n_1 = u_1\,, \qquad\qquad\qquad u_1 \sim \mathcal{N}(0,1)\,,$$
$$n_2 = -n_1 + \frac{1}{3}u_2\,, \qquad\qquad u_2 \sim \mathcal{N}(0,1)\,,$$
$$n_3 = \sin\left[0.1\pi(n_2 + 2.0)\right] + 0.2u_3\,, \qquad u_3 \sim \mathcal{N}(0,1)\,,$$
$$n_4 = \sin\left[0.25\pi(n_3 - n_1 + 2.0)\right] + 0.2u_4\,, \quad u_4 \sim \mathcal{N}(0,1)\,,$$

where there are four endogenous variables $(n_1, n_2, n_3, n_4)$ and four noise variables $(u_1, u_2, u_3, u_4)$. $n_1$ is the confounder of $n_2$ and $n_4$. $(n_2, n_3, n_4)$ is a chain, i.e., $n_2$ causes $n_3$, followed by $n_4$.

*Toy* 4.

$$
\begin{aligned}
n_1 &= u_1\,, & u_1 &\sim \mathcal{N}(0,1)\,, \\
n_2 &= -n_1 + \frac{1}{3}u_2\,, & u_2 &\sim \mathcal{N}(0,1)\,, \\
n_3 &= \sin\left[0.3\pi(n_2 + 2.0)\right] + 0.2u_3\,, & u_3 &\sim \mathcal{N}(0,1)\,,
\end{aligned}
$$

where there are three endogenous variables $(n_1, n_2, n_3)$ and three noise variables $(u_1, u_2, u_3)$. $(n_1, n_2, n_3)$ is a chain, i.e., $n_1$ causes $n_2$, followed by $n_3$.

Table 4: MAE Results on *Toy 1* to *Toy 4*. For simplicity, we use $do$ operator in the table to save room, and when natural counterfactuals are referred to, $do$ means $change$.

| Dataset | *Toy 1* | | *Toy 2* | | *Toy 3* | | | | | | *Toy 4* | | |
|---|---|---|---|---|---|---|---|---|---|---|---|---|---|
| $do$ or $change$ | $do(n_1)$ | | $do(n_2)$ | $do(n_1)$ | $do(n_1)$ | | | $do(n_2)$ | | $do(n_3)$ | $do(n_1)$ | | $do(n_2)$ |
| Outcome | $n_2$ | $n_3$ | $n_3$ | $n_2$ | $n_2$ | $n_3$ | $n_4$ | $n_3$ | $n_4$ | $n_4$ | $n_2$ | $n_3$ | $n_3$ |
| Non-backtracking | 0.477 | 0.382 | 0.297 | 0.315 | 0.488 | 0.472 | 0.436 | 0.488 | 0.230 | 0.179 | 0.166 | 0.446 | 0.429 |
| Ours | 0.434 | 0.354 | 0.114 | 0.303 | 0.443 | 0.451 | 0.423 | 0.127 | 0.136 | 0.137 | 0.158 | 0.443 | 0.327 |

In the main paper, we have explained experiments on *Toy 1* in detail. As shown in Table 4, our performance on *Toy 2-4* shows a big margin compared with non-backtracking counterfactuals since natural counterfactuals consistently make interventions feasible, while part of hard interventions may not be feasible in non-backtracking counterfactuals.

**Visualization Results on *Toy 1*.** In Fig. 3, larger figures are displayed, which are identical to those shown in Fig. 1, with the only difference being their size.

### A.2 MorphoMNIST

The MorphoMNIST comes from [25], where there are 60000 images as training set and 10,000 images as test dataset. Fig. 4 (a) shows the causal graph for generating MorphoMNIST; specifically, stroke thickness $t$ causes the brightness intensity $i$, and both thickness $t$ and intensity $i$ cause the digit $x$. Fig. 4 (b) show some samples from MorphoMNIST. The ground-truth SCM is as follows:

$$
\begin{aligned}
t &= 0.5 + u_t\,, & u_t &\sim \Gamma(10,5)\,, \\
i &= 191 \cdot \sigma(0.5 \cdot u_i + 2 \cdot t - 5) + 64\,, & u_i &\sim \mathcal{N}(0,1)\,, \\
x &= \text{SetIntensity}(\text{SetThickness}(u_x; t); i)\,, & u_x &\sim \text{MNIST}\,,
\end{aligned}
$$

where $u_t$, $u_i$, and $u_x$ are noise variables, and $\sigma$ is the sigmoid function. $\text{SetThickness}(\,\cdot\,; t)$ and $\text{SetIntensity}(\,\cdot\,; i)$ are the operations to set an MNIST digit $u_x$'s thickness and intensity to $i$ and $t$ respectively, and $x$ is the generated image.

Table 5: MorphoMNIST results of $change(i)$ or $do(i)$ using V-SCM

| Intersection between Ours and NB | | | (NC=1, NB=1) | (NC=1, NB=0) | (NC=0, NB=1) | (NC=0, NB=0) |
|---|---|---|---|---|---|---|
| Number of Intersection | | | 5865 | 3159 | 0 | 975 |
| Non-backtracking | $t$'s MAE | 0.283 | 0.159 | 0.460 | 0.000 | 0.450 |
| | $i$'s MAE | 6.56 | 3.97 | 8.95 | 0.000 | 14.3 |
| Ours | $t$'s MAE | 0.164 | 0.160 | 0.171 | 0.000 | 0.466 |
| | $i$'s MAE | 4.18 | 4.01 | 4.49 | 0.000 | 14.1 |

**Quantitative Results of $change(i)$ or $do(i)$.** We use V-SCM to do counterfactual task of $change(i)$ (where $\epsilon = 10^{-3}$) or $do(i)$ with multiple random seeds on test set. In Table 5, the first column shows

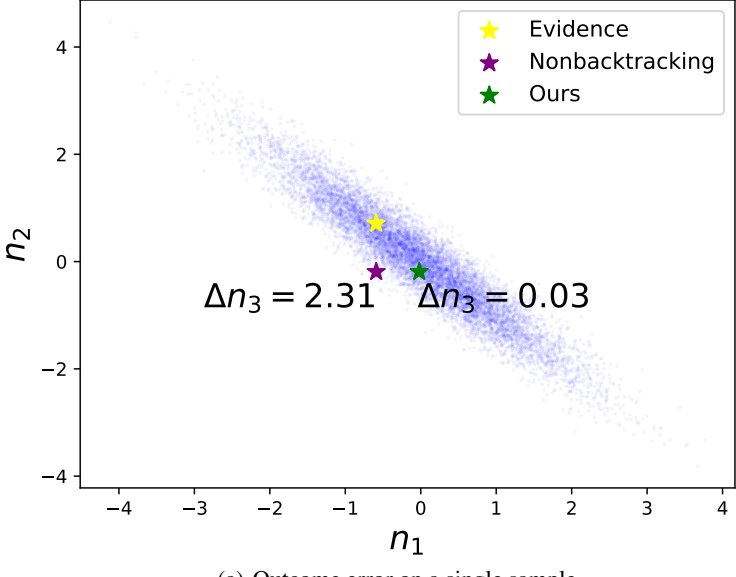

(a) Outcome error on a single sample

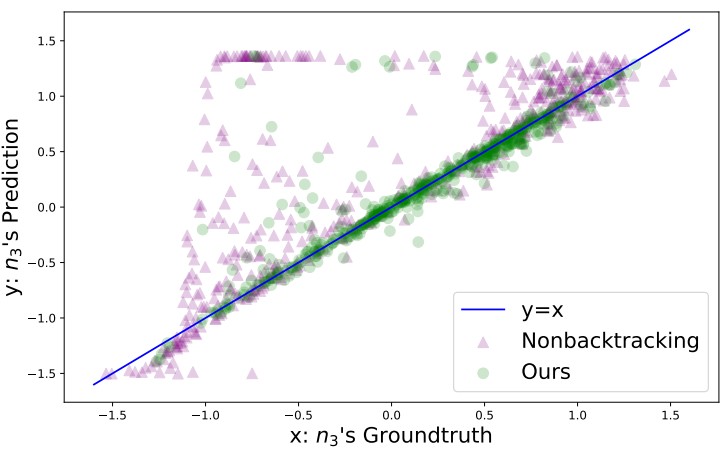

(b) Groundtruth-Prediction Scatter Plot

Figure 3: The Visualization Results on *Toy 1*.

the MAE of $(t, i)$, indicating our results outperform that of non-backtracking, since our approach consistently determines LBF interventions.

**The Effectiveness of FIO**. Next, we focus on the rest four-column results. In both types of counter-factuals, we use the same value $i$ in $do(i)$ and $change(i)$. We can calculate which image satisfies $\epsilon$-natural generation. In the table, "NC" indicates the set of counterfactuals after FIO. Notice that "NC" set does not mean the results of natural counterfactuals, since some results do still not satisfy $\epsilon$-natural generation after FIO. "NC=1" mean the set containing data points satisfying $\epsilon$-natural generation and "NC=0" contains data not satisfying $\epsilon$-natural generation after FIO. Similarly, "NB=1" means the set containing data points satisfying naturalness criteria in non-backtracking counterfactuals. (NC=1, NB=1) presents the intersection of "NC=1" and "NB=1". A similar logic is adopted for the other three combinations. The number of counterfactual data points is $10,000$ in two types of counterfactuals.

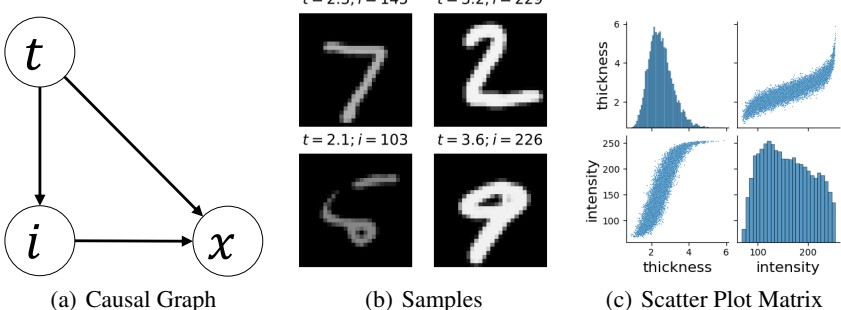

Figure 4: Causal Graph and samples of MorphoMNIST.

In (NC=1, NB=1) containing $5865$ data points, our performance is similar to the non-backtracking, since FIO does not do backtracking when hard interventions have satisfied $\epsilon$-natural generation. In (NC=1, NB=0), there are $3159$ data points, which are "unnatural" points in non-backtracking counterfactuals. After FIO, this huge amount of data points becomes "natural". Here, our approach significantly reduces errors, achieving a $62.8\%$ reduction in thickness $t$ and $49.8\%$ in intensity $i$, the most substantial improvement among the four sets in Table 5. The number of points in (NC=0, NB=1) is zero, showing the stability of our algorithm since our FIO framework will change the hard, feasible intervention into an unfeasible intervention. Two types of counterfactuals perform similarly in the set (NC=0, NB=0), also showing the stability of our approach.

Table 6: Unfeasible solutions per 10,000 instances on MorphoMNIST

| $\epsilon$ | Unfeasible Solutions |
|---|---|
| 1e-4 | 794 |
| 1e-3 | 975 |
| 1e-2 | 1166 |

**Ablation Study on $\epsilon$.** In Table 6 provided, we report the frequency of unfeasible solutions per 10,000 instances on MorphoMNIST, following optimization within the V-SCM framework. The data reveals a consistent trend: as the value of $\epsilon$ increases, the frequency of unfeasible solutions also rises. This pattern occurs because a higher $\epsilon$ corresponds to a stricter standard of naturalness, making it more challenging to achieve feasible outcomes.

**Visualization of Counterfactual Images.** Fig. 5 shows counterfactual images (second row), based on the evidence images (first row), with intended changes on $i$. The third row illustrates the differences between evidence and counterfactual images. Focusing on the first counterfactual image from non-backtracking and natural counterfactuals respectively, in non-backtracking, despite $do(i)$ where thickness value $4.2$ should remain unchanged, the counterfactual image shows reduced thickness, consistent with the measured counterfactual thickness of $2.6$. In contrast, natural counterfactuals yield an estimated counterfactual thickness ($t^*$ in MS) closely matching the original counterfactual thickness ($t^*$ in CF), due to backtracking for an LBF intervention on the earlier causal variable $t$, thereby maintaining $(t, i)$ within the data distribution. Observing other images also shows larger errors in non-backtracking counterfactual images.

### A.3  3DIdentBOX

The *3DIdentBOX* datasets, first introduced in [4], come with official code for generating customized versions of these datasets. They consist of images created with Blender, each depicting a teapot with seven attributes, such as position, rotation, and hue, determined by seven ground-truth variables.

In our experiment with the 3DIdentBOX, which comprises six datasets, we focus on the *positions-rotations-hue* dataset. We expand this into two datasets, Weak-3DIdent and Strong-3DIdent. Each dataset includes seven variables, besides the image variable $x$, with specifics outlined in Table 7. Every image features a teapot, with variables categorized into three groups: positions $(x, y, z)$,

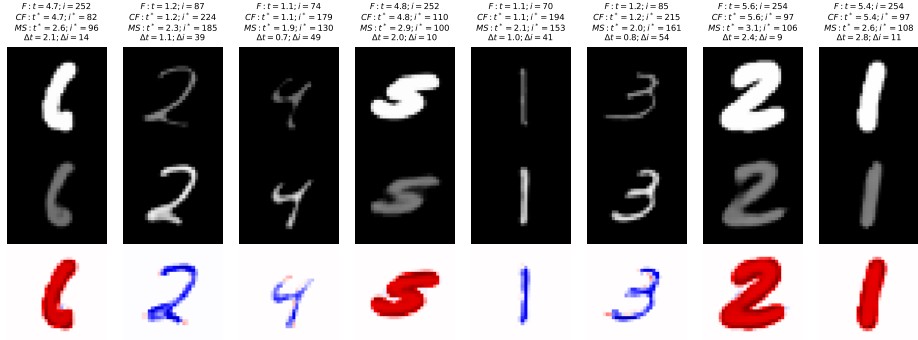

(a) Results of Non-backtracking Counterfactuals

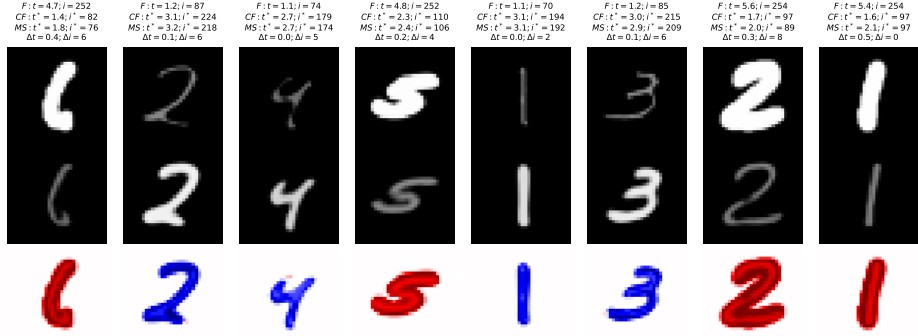

(b) Results of Natural Counterfactuals

Figure 5: Visualization Results on MorphoMNIST: "F" stands for factual values, "CF" for counterfactual values, and "MS" for estimated counterfactual values of $(t, i)$. $(\Delta t, \Delta i)$ represents the absolute errors between counterfactual and estimated counterfactual values of $(t, i)$.

Table 7: Details of variables in 3DIndentBOX. Object refers to the teapot in each image. The support of each variable is $[-1, 1]$. The real visual range are listed in the column **Visual Range**.

| Information Block | Variables | Support | Description | Visual Range |
|---|---|---|---|---|
| Position | x | [-1, 1] | Object x-coordinate | - |
| | y | [-1, 1] | Object y-coordinate | - |
| | z | [-1, 1] | Object z-coordinate | - |
| Rotation | $\gamma$ | [-1, 1] | Spotlight rotation angle | $[0°, 360°]$ |
| | $\alpha$ | [-1, 1] | Object $\alpha$-rotation angle | $[0°, 360°]$ |
| | $\beta$ | [-1, 1] | Object $\beta$-rotation angle | $[0°, 360°]$ |
| Hue | b | [-1, 1] | Background HSV color | $[0°, 360°]$ |

rotations $(\gamma, \alpha, \beta)$, and hue $b$, representing seven teapot attributes, as depicted in 7 (a). Fig. 7 (b) illustrates that both datasets share the same causal graph. It is important to note a distinction between Weak-3DIdent and Strong-3DIdent. In Weak-3DIdent, there exists a weak causal relationship between the variables of each pair, as shown in Fig. 7 (c), whereas in Strong-3DIdent, the causal relationship is stronger, as depicted in Fig. 7 (d). The distributions of several parent variables of image $x$ in these datasets are detailed in Table 8.

**Visualization on Strong-3DIdent.** Fig. 6 displays counterfactuals, with the text above the evidence images (first row) indicating errors for the counterfactual images (second row). In Fig. 6 (a), it is evident that some images (second, third, and fifth images in particular), generated by non-backtracking counterfactuals are less recognizable and have larger errors. Conversely, our counterfactual images exhibit better visual clarity and more distinct shapes, as our natural counterfactuals consistently ensure feasible interventions, resulting in more natural-looking images.

Table 8: Distributions in Weak-3DIdent and Strong-3DIdent. $\mathcal{N}_{wt}(y, 1)$ refers to a normal distribution truncated to the interval $[-1, 1]$ and $\mathcal{N}_{st}(y, 1)$ means a normal distribution truncated to the interval $[\min(1, y + 0.2), \max(-1, y - 0.2)]$, where $\min$ and $\max$ indicate operations that select smaller and bigger values respectively. $\mathcal{N}_{wt}(\alpha, 1)$ and $\mathcal{N}_{st}(\alpha, 1)$ are identical to $\mathcal{N}_{wt}(y, 1)$ and $\mathcal{N}_{st}(y, 1)$ respectively. $U$ refers to uniform distribution.

| Variables | Weak-3DIdent Distribution | Strong-3DIdent Distribution |
|---|---|---|
| $c = (x, y, z)$ | $c \sim (\mathcal{N}_{wt}(y, 1), U(-1, 1), U(-1, 1))$ | $c \sim (\mathcal{N}_{st}(y, 1), U(-1, 1), U(-1, 1))$ |
| $s = (\gamma, \alpha, \beta)$ | $s \sim (\mathcal{N}_{wt}(\alpha, 1), U(-1, 1), \mathcal{N}_{w}t(z, 1))$ | $s \sim (\mathcal{N}_{st}(\alpha, 1), U(-1, 1), \mathcal{N}_{st}(z, 1))$ |
| $b$ | $b \sim U(-1, 1)$ | $b \sim U(-1, 1)$ |

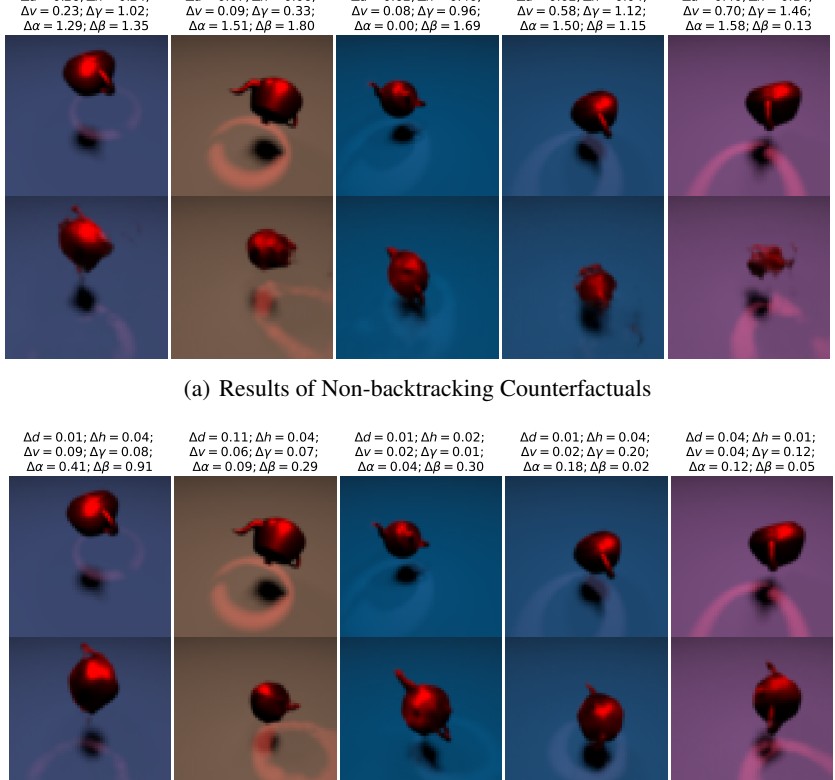

(a) Results of Non-backtracking Counterfactuals

(b) Results of Natural Counterfactuals

Figure 6: Visualization Results on Stong-3DIdent.

## A.4 Standard Deviation of Experimental Results

This section presents the standard deviation of all experimental results, demonstrating that the standard deviation for our natural counterfactuals is generally lower. This indicates the increased reliability of our approach, achieved by necessary backtracking to ensure counterfactuals remain within data distributions.

Table 9: Standard Deviation of Results on *Toy 1* to *Toy 4*.

| Dataset | Toy 1 | | | Toy 2 | Toy 3 | | | | | | Toy 4 | | |
|---|---|---|---|---|---|---|---|---|---|---|---|---|---|
| *do* or *change* | $do(n_1)$ | | $do(n_2)$ | $do(n_1)$ | $do(n_1)$ | | | $do(n_2)$ | | $do(n_3)$ | $do(n_1)$ | | $do(n_2)$ |
| Outcome | $n_2$ | $n_3$ | $n_3$ | $n_2$ | $n_2$ | $n_3$ | $n_4$ | $n_3$ | $n_4$ | $n_4$ | $n_2$ | $n_3$ | $n_3$ |
| Non-backtracking | 0.00184 | 0.00628 | 0.00432 | 0.00164 | 0.00448 | 0.00686 | 0.00495 | 0.0112 | 0.00556 | 0.00142 | 0.000514 | 0.00623 | 0.00238 |
| Ours | 0.00409 | 0.00684 | 0.00295 | 0.00191 | 0.00116 | 0.00461 | 0.00201 | 0.00504 | 0.00531 | 0.00155 | 0.000235 | 0.00518 | 0.00143 |

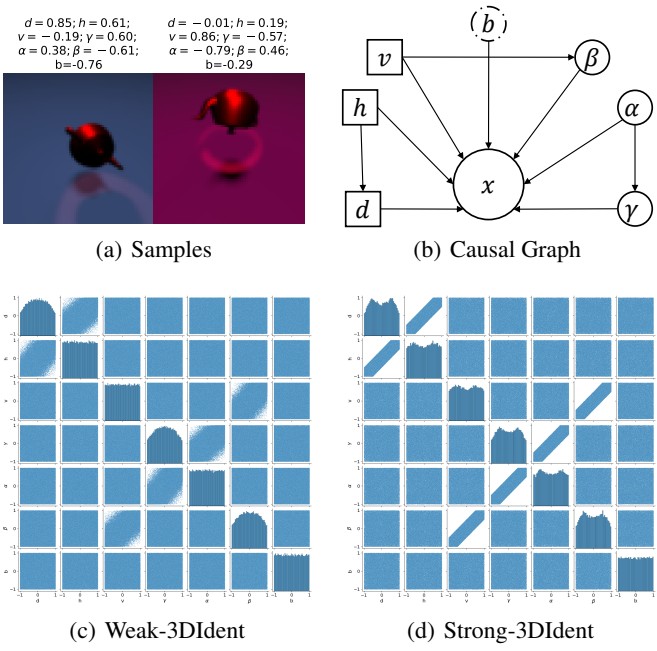

(a) Samples        (b) Causal Graph

(c) Weak-3DIdent        (d) Strong-3DIdent

Figure 7: Samples, Causal Graph, Scatter Plot Matrices of Weak-3DIdent and Strong-3DIdent.

Table 10: Standard Deviation of Results on MorphoMNIST

| Intersection between Ours and NB | | | (NC=1, NB=1) | (NC=1, NB=0) | (NC=0, NB=1) | (NC=0, NB=0) |
|---|---|---|---|---|---|---|
| Number of Intersection | | | 39.84 | 81.43 | 0.00 | 54.14 |
| Non-backtracking | $t$'s MAE | 0.00322 | 0.00172 | 0.00670 | 0.000 | 0.0178 |
| | $i$'s MAE | 0.0496 | 0.0596 | 0.0508 | 0.000 | 0.110 |
| Ours | $t$'s MAE | 0.00137 | 0.00222 | 0.00157 | 0.000 | 0.0149 |
| | $i$'s MAE | 0.0359 | 0.0551 | 0.0157 | 0.000 | 0.0853 |

Table 11: Standard Deviation of Ablation Study on $\epsilon$ Using MorphoMNIST

| Model | $\epsilon$ | CFs | do($t$) $t$ | $i$ | do($i$) $t$ | $i$ |
|---|---|---|---|---|---|---|
| V-SCM | - | NB | 0.000512 | 0.0172 | 0.00322 | 0.0496 |
| | $10^{-4}$ | | 0.00159 | 0.0210 | 0.00183 | 0.0561 |
| | $10^{-3}$ | Ours | 0.00124 | 0.0217 | 0.00137 | 0.0359 |
| | $10^{-2}$ | | 0.000954 | 0.0382 | 0.000868 | 0.0556 |
| H-SCM | - | NB | 0.000915 | 0.0229 | 0.000832 | 0.0245 |
| | $10^{-4}$ | | 0.000920 | 0.0178 | 0.000922 | 0.0138 |
| | $10^{-3}$ | Ours | 0.000611 | 0.0206 | 0.000289 | 0.0264 |
| | $10^{-2}$ | | 0.000787 | 0.0244 | 0.000431 | 0.0258 |

Table 12: Standard Deviation of Results on Weak-3DIdent and Stong-3DIdent

| Dataset | Counterfactuals | $d$ | $h$ | $v$ | $\gamma$ | $\alpha$ | $\beta$ | $b$ |
|---|---|---|---|---|---|---|---|---|
| Weak-3DIdent | Non-backtracking | 3.68e-05 | 0.000133 | 0.000226 | 0.00422 | 0.00310 | 0.00357 | 1.29e-05 |
| | Ours | 4.27e-05 | 7.22e-05 | 0.000249 | 0.00558 | 0.00278 | 0.00136 | 3.33e-05 |
| Stong-3DIdent | Non-backtracking | 0.00233 | 0.000864 | 0.00127 | 0.00933 | 0.00307 | 0.00452 | 1.49e-05 |
| | Ours | 0.00166 | 0.000774 | 0.000229 | 0.00908 | 0.00955 | 0.00816 | 2.97e-05 |

# B  Proof for Theorem 4.1

Theorem 4.1 shows that our natural counterfactuals are identifiable given certain conditions. We unitize Theorem 1 from [20] to assist us to prove it, shown as below.

**Theorem B.1** (Identifiable Counterfactuals). *Given actual distribution* $p(\mathbf{PA}_i, \mathbf{U}_i)$, *Suppose* $\mathbf{V}_i$ *satisfies the following function:*

$$\mathbf{V}_i := f_i(\mathbf{PA}_i, \mathbf{U}_i)$$

*where* $\mathbf{U}_i \perp \mathbf{PA}_i$ *and assume unknown* $f_i$ *is smooth and strictly monotonic w.r.t.* $\mathbf{U}_i$ *for fixed values of* $\mathbf{PA}_i$. *If we have observed* $\mathbf{V}_i = v_i$ *and* $\mathbf{PA}_i = pa_i$, *with an intervention* $do(\mathbf{PA}_i = pa_i^*)$, *where* $pa_i^*$ *is within the support of* $p(\mathbf{PA}_i)$, *the counterfactual outcome is identifiable:*

$$\mathbf{V}_i | do(\mathbf{PA}_i = pa_i^*), \mathbf{V}_i = v_i, \mathbf{PA}_i = pa_i \tag{7}$$

The theorem is not identical to Theorem 1 from [20], as we incorporate additional assumptions from [20] in our proof to express it strictly. The proof is as below.

*Proof.* **Step 1. Identifiability of** $\mathbf{AN}(\mathbf{C})$. Given an intervention $do(\mathbf{C} = \mathbf{c}^*)$ on the set of variables $\mathbf{C}$, where $\mathbf{AN}(\mathbf{C})$ denotes the ancestors of $\mathbf{C}$ including $\mathbf{C}$ itself. By definition of the intervention and the causal graph structure, the value of $\mathbf{AN}(\mathbf{C})$ under the intervention $do(\mathbf{C} = \mathbf{c}^*)$ can be uniquely determined. This is because the values of ancestors are not influenced by the intervention, while the values of $\mathbf{C}$ are directly determined by the intervention, making them identifiable.

**Step 2. Identifiability of** $\mathbf{V} \setminus \mathbf{AN}(\mathbf{C})$. There are two subsets in $\mathbf{V} \setminus \mathbf{AN}(\mathbf{C})$. (1) **ND**: Variables in $\mathbf{V} \setminus \mathbf{AN}(\mathbf{C})$ that are **n**ot **d**escendants of $\mathbf{AN}(\mathbf{C})$. These variables retain their values from the actual observed data since they are not influenced by the changes in $\mathbf{AN}(\mathbf{C})$ due to the causal independence. (2) **DS**: Variables in $\mathbf{V} \setminus \mathbf{AN}(\mathbf{C})$ that are descendants of $\mathbf{AN}(\mathbf{C})$. The values of these variables might change as a result of the intervention due to their causal dependency on $\mathbf{AN}(\mathbf{C})$.

*Identifiability of* **ND**. The counterfactual values of variables in **ND** remain as their observed values in the actual data, as they are causally independent of $\mathbf{AN}(\mathbf{C})$. Thus, their values are trivially identifiable.

*Identifiability of* **DS**. The counterfactual values of variables in **DS** depend on the values of their parents. For variable set $\mathbf{DS}_1$ in **DS** whose parents only come from $\mathbf{ND} \cup \mathbf{AN}(\mathbf{C})$, Their counterfactual values are identifiable by Theorem B.1, based on the fact that the counterfactual values of their parent variables are identifiable and lie within the observed distribution. Now, the the set of identifiable variables is $\mathbf{DS}_1 \cup \mathbf{ND} \cup \mathbf{AN}(\mathbf{C})$. This allows us to recursively expand the set of identifiable variables. By incorporating variables from **DS**, whose counterfactual values are identifiable, into the identifiable variable set $\mathbf{ND} \cup \mathbf{AN}(\mathbf{C})$, this set is iteratively enlarged. This process is repeated until all variables in $\mathbf{V}$ are included in the set of identifiable variables, thus establishing the identifiability of their values under the counterfactual scenario induced by the intervention $do(\mathbf{C} = \mathbf{c}^*)$.

**Conclusion.** By proving the identifiability of both $\mathbf{AN}(\mathbf{C})$ and all variables in $\mathbf{V} \setminus \mathbf{AN}(\mathbf{C})$ (partitioned into **ND** and **DS**), we establish that the entire set of endogenous variables $\mathbf{V}$ is identifiable under the intervention $do(\mathbf{C} = \mathbf{c}^*)$. **This implies that the values of all variables in $\mathbf{V}$ can be uniquely determined given the causal graph, the data distribution of endogenous (observed) variables, full evidence, and a single LBF intervention.**

$\square$

# C  Causal Model Training

Our study focuses on counterfactual inference and we directly use two state-of-the-art deep-learning SCM models to learn conditional distributions among variables using a dataset, i.e., V-SCM [25] and H-SCM [28]. Specifically, we use code of [28] containing the implementation of V-SCM and H-SCM. Take MorphoMNIST as an example, in both two models, normalizing flows are firstly trained to learn causal mechanisms for all variables except image $x$, i.e., $(t, i)$, and a conditional VAE is used to model image $x$ given its parents $(t, i)$. For V-SCM, the conditional VAE uses a normal VAE framework, while H-SCM uses a hierarchical VAE structure [21] to better capture the distribution of images.

**Toy Experiments.** In the case of four toy experiments, we exclusively employed normalizing flows due to the fact that all variables are one-dimensional. Our training regimen for the flow-based model spanned 2000 epochs, utilizing a batch size of 100 in conjunction with the AdamW optimizer [19]. We initialized the learning rate to $10^{-3}$, set $\beta_1$ to 0.9, $\beta_2$ to 0.9.

**MorphoMNIST.** We first train normalized flows to learn causal mechanisms of thickness and intensity $(t, i)$. Other hyper-parameters are similar to those of toy experiments. Then, we train two VAE-based models (V-SCM and H-SCM) to learn $x$ given $(t, i)$ respectively. The architectures of the two models are identical to [28]. V-SCM and H-SCM underwent training for 160 epochs. We employed a batch size of 32 and utilized the AdamW optimizer. The initial learning rate was set to $1e-3$ and underwent a linear warmup consisting of 100 steps. We set $\beta_1$ to 0.9, $\beta_2$ to 0.9, and applied a weight decay of 0.01. Furthermore, we implemented gradient clipping at a threshold of 350 and introduced a gradient update skipping mechanism, with a threshold set at 500 based on the $L^2$ norm. During the testing, i.e., counterfactual inference, we test performance on both models respectively, with the normalized flows.

3**DIdentBOX.** Similar to experiments on MorphoMNIST, we first train normalized flows. Compared with V-SCM, H-SCM is more powerful to model complex data like 3DIdentBOX, of which the size of the image is $64 \times 64 \times 4$. Then, we train H-SCM to capture the distribution of $x$ given its parents for 500 epochs with a batch size of 32. The hyper-parameters are the same as experiments on MorphoMNIST.

All the experiments above were run on NVIDIA RTX 4090 GPUs.

## D  Feasible Intervention Optimization

To implement a particular method, we plug the distance measure (Eqn. 4) and a naturalness constraint (we use Choice (3) in Sec. 4.1.1 in the experiments) into Eqn. 2 of the FIO framework:

$$
\begin{aligned}
&\underset{an(\mathbf{A})^*}{\text{minimize}} \ \|an(\mathbf{A})^* - an(\mathbf{A})\|_1 \\
&s.t. \quad \mathbf{A} = \mathbf{a}^*, \\
&\quad \epsilon < F(\mathbf{V}_j = \mathbf{v}_j^* | pa_j^*) < 1 - \epsilon, \forall \mathbf{V}_j \in \mathbf{AN}(\mathbf{A}).
\end{aligned}
\tag{8}
$$

According to the properties of reversible functions in the causal model, endogenous value $an(\mathbf{A})^*$ is reversible to noise values $\mathbf{u}^*_{\mathbf{AN}(\mathbf{A})}$ of $\mathbf{AN}(\mathbf{A})^*$'s corresponding noise variables. Hence, optimizing endogenous value is equivalent to optimizing noise value. Then we have:

$$
\begin{aligned}
&\underset{\mathbf{u}^*_{\mathbf{AN}(\mathbf{A})}}{\text{minimize}} \ \sum_{\mathbf{u}_j^* \in \mathbf{u}^*_{\mathbf{AN}(\mathbf{A})}} |\hat{f}(pa_{\mathbf{A}}^*, \mathbf{u}_j^*) - \mathbf{v}_j| \\
&s.t. \quad \mathbf{u}_{\mathbf{A}}^* = \hat{f}_{\mathbf{A}}^{-1}(pa_{\mathbf{A}}^*, \mathbf{a}^*), \\
&\quad \epsilon < F(\mathbf{u}_j^*) < 1 - \epsilon, \forall \mathbf{U}_j \in \mathbf{U}_{\mathbf{AN}(\mathbf{A})}.
\end{aligned}
\tag{9}
$$

We finally use the Lagrangian method [5] to optimize our objective loss to get the counterfactual value given actual value and expected change $\mathbf{A} = \mathbf{a}^*$ as below:

$$
\begin{aligned}
&\mathcal{L}(\mathbf{u}^*_{\mathbf{AN}(\mathbf{A})}) = \sum_{\mathbf{u}_j^* \in \mathbf{u}^*_{\mathbf{AN}(\mathbf{A})}} |\hat{f}_j(\mathbf{u}_j^*, pa_j^*) - \mathbf{v}_j| + w_\epsilon \sum_{\mathbf{u}_j^* \in \mathbf{u}^*_{\mathbf{AN}(\mathbf{A})}} [\max(\epsilon - F(\mathbf{u}_j^*), 0) + \max(\epsilon + F(\mathbf{u}_j^*) - 1, 0)] \\
&s.t. \quad \mathbf{u}_{\mathbf{A}}^* = \hat{f}_{\mathbf{A}}^{-1}(pa_{\mathbf{A}}^*, \mathbf{a}^*)
\end{aligned}
\tag{10}
$$

where the optimization parameters are the counterfactual noise values of $\mathbf{A}$'s ancestors, $\mathbf{u}^*_{\mathbf{AN}(\mathbf{A})}$, and the function $\max(\cdot)$ returns the maximum value between two given values. The first term represents the measure of distance between two distinct worlds, while the second term enforces the constraint of $\epsilon$-natural generation. Here, the constant hyperparameter $w_\epsilon$ serves to penalize noise values situated in the tails of noise distributions. For simplicity, we use $\mathbf{A}$ as a subscript as an indicator of terms related to $\mathbf{A}$, instead of a number subscript. Notice that, in order to ensure hard constraint $\mathbf{A} = \mathbf{a}^*$, $\mathbf{A}$'s noise value $\mathbf{u}_{\mathbf{A}}^*$ is not optimized explicitly, since the value $\mathbf{pa}_{\mathbf{A}}^*$ is fully determined by $\mathbf{a}^*$ and other noise values.

**Hyper-parameters for Optimization and Judgment for Non-solution Cases.** The loss's parameter is thus $\mathbf{u}^*_{\mathbf{AN}}$, which fully determines the value $an(\mathbf{A})^*$ using the pretrained causal model, as explained

in Sec. C. In all experiments, we optimized $\mathbf{u}^*_{\mathbf{AN}}$ using the AdamW optimizer at a learning rate of $10^{-3}$ for $50,000$ steps. This approach's effectiveness is validated by the MorphoMNIST experiments, as shown in Table 5. **As $\hat{f}$ is not perfectly invertible in practice, it is important to ensure that $|pa^*_{\mathbf{A}} - \hat{f}_{\mathbf{A}}(\mathbf{u}^*_{\mathbf{A}}, \mathbf{a}^*)|$ (where $\mathbf{u}^*_{\mathbf{A}} = \hat{f}^{-1}_{\mathbf{A}}(pa^*_{\mathbf{A}}, \mathbf{a}^*)$) remains sufficiently small, ideally on the order of $10^{-8}$ according to our experience, to satisfy the constraint condition in Eqn. 10. In our experience, this deviation $|pa^*_{\mathbf{A}} - \hat{f}_{\mathbf{A}}(\mathbf{u}^*_{\mathbf{A}}, \mathbf{a}^*)|$ is typically sufficiently small throughout most of the optimization process. If not, please adjust hyperparameters, such as the learning rate and $w_\epsilon$, to achieve the desired accuracy.** Assuming $|pa^*_{\mathbf{A}} - \hat{f}_{\mathbf{A}}(\mathbf{u}^*_{\mathbf{A}}, \mathbf{a}^*)|$ is sufficiently small after optimization, if $\sum_{\mathbf{u}^*_j \in \mathbf{u}^*_{\mathbf{AN(A)}}} [\max(\epsilon - F(\mathbf{u}^*_j), 0) + \max(\epsilon + F(\mathbf{u}^*_j) - 1, 0)] > 0$ holds, this indicates that the case does not have a solution.

**Computational Complexity of FIO.** The complexity scales linearly with both the size of the causal graph, specifically the number of ancestors of $A$, and the dataset size. Formally, for one counterfactual query, the overall complexity is $O(KPT)$, where $K$ is the number of ancestors of $A$, $P$ is the number of parameters in the neural networks, and $T$ is the number of optimization steps. If we use each instance of a whole dataset as evidence and answer one counterfactual query for each instance, the overall complexity becomes $O(KPTM)$, where $M$ is the number of data points in the dataset. This linear scaling indicates that FIO is reasonably scalable, making it suitable for large causal graphs and datasets.

# E Differences between Natural Counterfactuals and Non-Backtracking Counterfactuals [26] or Prior-Based Backtracking Counterfactuals [35]

## E.1 Differences between Non-backtracking Counterfactuals and Ours

Non-backtracking counterfactuals only do a direct intervention on $\mathbf{A}$, while our natural counterfactuals do backtracking when the direct intervention is infeasible. Notice that when the direct intervention on $\mathbf{A}$ is already feasible, our procedure of natural counterfactuals will be automatically distilled to the non-backtracking counterfactuals. In this sense, non-backtracking counterfactual reasoning is our special case.

## E.2 Differences between Prior-Based Backtracking Counterfactuals and Ours

### (1) Intervention Approach and Resulting Changes:

Prior-based Backtracking Counterfactuals: These counterfactuals directly intervene on noise/exogenous variables, which can lead to unnecessary changes in the counterfactual world.[6] Consequently, the similarity between the actual data point and its counterfactual counterpart tends to be lower. In short, prior-based backtracking counterfactuals may introduce changes that are not needed.

Natural Counterfactuals: In contrast, our natural counterfactuals only engage in necessary backtracking when direct intervention is infeasible. This approach aims to ensure that the counterfactual world results from minimal alterations, maintaining a higher degree of fidelity to the actual world.

### (2) Counterfactual Worlds:

Prior-based Backtracking Counterfactuals: This approach assigns varying weights to the numerous potential counterfactual worlds capable of effecting the desired change. The weight assigned to each world is directly proportional to its similarity to the actual world. it is worth noting that among this array of counterfactual worlds, some may exhibit minimal resemblance to the actual world, even when equipped with complete evidence, including the values of all endogenous variables. This

---

[6] To avoid misleading readers, we clarify why we apply the concept of intervention to exogenous variables. Prior-based backtracking counterfactuals do not appeal to interventions in the usual Pearlian sense, for those are restricted to endogenous variables and are associated with breaking endogenous mechanisms. However, formally speaking, changing the values of exogenous variables is analogous to intervening on exogenous variables, in the sense that they change those variables without affecting the mechanisms for other variables in the model (the invariance of the mechanisms for other variables is for many the crucial hallmark of an intervention). In fact, some authors explicitly advocate applying the notion of intervention to exogenous variables as well [33].

divergence arises because, by sampling from the posterior distribution of exogenous variables, even highly dissimilar worlds may still be drawn.

Natural Counterfactuals: In contrast, our natural counterfactuals prioritize the construction of counterfactual worlds that closely emulate the characteristics of the actual world through an optimization process. As a result, in most instances, one actual world corresponds to a single counterfactual world when employing natural counterfactuals with full evidence.

**(3) Implementation Practicality:**

Prior-based Backtracking Counterfactuals: The practical implementation of prior-based backtracking counterfactuals can be a daunting challenge. To date, we have been prevented from conducting a comparative experiment with this approach due to uncertainty about its feasibility in practical applications. Among other tasks, the computation of the posterior distribution of exogenous variables can be a computationally intensive endeavor. Furthermore, it is worth noting that the paper [35] provides only rudimentary examples without presenting a comprehensive algorithm or accompanying experimental results.

Natural Counterfactuals: In stark contrast, our natural counterfactuals have been meticulously designed with practicality at the forefront. We have developed a user-friendly algorithm that can be applied in real-world scenarios. Rigorous experimentation, involving four simulation datasets and two public datasets, has confirmed the efficacy and reliability of our approach. This extensive validation underscores the accessibility and utility of our algorithm for tackling specific problems, making it a valuable tool for practical applications.

# F  Observations about the Prior-Based Backtracking Counterfactuals [35]

## F.1  Possibility of Gratuitous Changes

A theory of backtracking counterfactuals was recently proposed by [35], which utilizes a prior distribution $p(\mathbf{U}, \mathbf{U}^*)$ to establish a connection between the actual model and the counterfactual model. This approach allows for the generation of counterfactual results under any condition by considering paths that backtrack to exogenous noises and measuring closeness in terms of noise terms. As a result, for any given values of $\mathbf{E} = \mathbf{e}$ and $\mathbf{A}^* = \mathbf{a}^*$, it is possible to find a sampled value $(\mathbf{U} = \mathbf{u}, \mathbf{U}^* = \mathbf{u}^*)$ from $p(\mathbf{U}, \mathbf{U}^*)$ such that $\mathbf{E}_{\mathcal{M}(\mathbf{u})} = \mathbf{e}$ and $\mathbf{A}^*_{\mathcal{M}^*(\mathbf{u}^*)} = \mathbf{a}^*$, as described in [35]. This holds true even in cases where $\mathbf{V} \setminus \mathbf{E} = \emptyset$ and $\mathbf{V}^* \setminus \mathbf{A}^* = \emptyset$, implying that any combination of endogenous values $\mathbf{E} = \mathbf{e}$ and $\mathbf{A}^* = \mathbf{a}^*$ can co-occur in the actual world and the counterfactual world, respectively. In essence, there always exists a path $(\mathbf{v} \to \mathbf{u} \to \mathbf{u}^* \to \mathbf{v}^*)$ that connects $\mathbf{V} = \mathbf{v}$ and $\mathbf{V}^* = \mathbf{v}^*$ through a value $(\mathbf{U} = \mathbf{u}, \mathbf{U}^* = \mathbf{u}^*)$, where $\mathbf{v}$ and $\mathbf{v}^*$ represent any values sampled from $p_{\mathcal{M}}(\mathbf{V})$ and $p_{\mathcal{M}^*}(\mathbf{V}^*)$, respectively.

However, thanks to this feature, this understanding of counterfactuals may allow for what appears to be gratuitous changes in realizing a counterfactual supposition. This occurs when there exists a value assignment $\mathbf{U}^* = \mathbf{u}^*$ that satisfies $\mathbf{E}^*_{\mathcal{M}^*(\mathbf{u}^*)} = \mathbf{e}$ and $\mathbf{A}^*_{\mathcal{M}^*(\mathbf{u}^*)} = \mathbf{a}^*$ in the same world. In such a case, intuitively we ought to expect that $\mathbf{E}^* = \mathbf{e}$ should be maintained in the counterfactual world (as in the factual one). However, there is in general a positive probability for $\mathbf{E}^* \neq \mathbf{e}$. This is due to the existence of at least one "path" from $\mathbf{E} = \mathbf{e}$ to any value $\mathbf{v}^*$ sampled from $p_{\mathcal{M}^*}(\mathbf{V}^*|\mathbf{A}^* = \mathbf{a}^*)$ by means of at least one value $(\mathbf{U} = \mathbf{u}, \mathbf{U}^* = \mathbf{u}^*)$, allowing $\mathbf{E}^*$ to take any value in the support of $p_{\mathcal{M}^*}(\mathbf{E}^*|\mathbf{A}^* = \mathbf{a}^*)$.

In the case where $\mathbf{A}^* = \emptyset$, an interesting observation is that $\mathbf{E}^*$ can take any value within the support of $p_{\mathcal{M}^*}(\mathbf{E}^*)$. Furthermore, when examining the updated exogenous distribution, we find that in Pearl's non-backtracking framework, it is given by $p_{\mathcal{M}^*}(\mathbf{U}^*|\mathbf{E}^* = \mathbf{e})$. However, in [35]'s backtracking framework, the updated exogenous distribution becomes $p_B(\mathbf{U}^*|\mathbf{E} = \mathbf{e}) = \int p(\mathbf{U}^*|\mathbf{U})p_{\mathcal{M}}(\mathbf{U}|\mathbf{E} = \mathbf{e})d(\mathbf{U}) \neq p_{\mathcal{M}^*}(\mathbf{U}^*|\mathbf{E}^* = \mathbf{e})$, since using $\mathbf{u}^*$ sampled from $p(\mathbf{U}^*|\mathbf{U} = \mathbf{u})$ (where $\mathbf{u}$ is any value of $\mathbf{U}$) can result in any value of all endogenous variables $\mathbf{V}^*$. Therefore, [35]'s backtracking counterfactual does not reduce to Pearl's counterfactual even when $\mathbf{A}^* = \emptyset$.

## F.2 Issues with the Distance Measure

In Equation 3.16 of [35], Mahalanobis distance is used for real-valued $\mathbf{U} \in \mathbb{R}^m$, defined as $d(\mathbf{u}^*, \mathbf{u}) = \frac{1}{2}(\mathbf{u}^* - \mathbf{u})^{\mathrm{T}}\Sigma^{-1}(\mathbf{u}^* - \mathbf{u})$. However, it should be noted that the exogenous variables are not identifiable. There are several issues with using the Mahalanobis distance in this context.

Firstly, selecting different exogenous distributions would result in different distances. This lack of identifiability makes the distance measure sensitive to the choice of exogenous distributions.

Secondly, different noise variables may have different scales. By using the Mahalanobis distance, the variables with larger scales would dominate the distribution changes, which may not accurately reflect the changes in each variable fairly.

Thirdly, even if the Mahalanobis distance $d(\mathbf{u}^*, \mathbf{u})$ is very close to 0, it does not guarantee that the values of the endogenous variables are similar. This means that the Mahalanobis distance alone may not capture the similarity or dissimilarity of the endogenous variables adequately.

# G  Another Type of Minimal Change: Minimal Change in Local Causal Mechanisms.

Changes in local mechanisms are the price we pay to do interventions, since interventions are from outside the model and sometimes are imposed on a model by us. Hence, we consider the minimal change in local causal mechanisms in $\mathbf{A}$'s ancestor set $\mathbf{AN}(\mathbf{A})$. With $L^1$ norm, the total distance of mechanisms in $\mathbf{AN}(\mathbf{A})$, called **mechanism distance**, is defined as:

$$D(\mathbf{u}_{an(\mathbf{A})}, \mathbf{u}_{an(\mathbf{A})^*}) = \sum_{\mathbf{u}_j \in \mathbf{u}_{an(\mathbf{A})}, \mathbf{u}_j^* \in \mathbf{u}_{an(\mathbf{A})^*}} w_j |F(\mathbf{u}_j) - F(\mathbf{u}_j^*)| \tag{11}$$

where $\mathbf{u}_{an(\mathbf{A})}$ is the value of $\mathbf{A}$'s exogenous ancestor set $\mathbf{U}_{\mathbf{AN}(\mathbf{A})}$ when $\mathbf{AN}(\mathbf{A}) = an(\mathbf{A})$ in the actual world. $\mathbf{u}_{an(\mathbf{A})^*}$ is the value of $\mathbf{U}_{\mathbf{AN}(\mathbf{A})}$ when $\mathbf{AN}(\mathbf{A}) = an(\mathbf{A})^*$ in the counterfactual world. $D(\mathbf{u}_{an(\mathbf{A})}, \mathbf{u}_{an(\mathbf{A})^*})$ represents the distance between actual world and counterfactual world. $w_j$ represents a fixed weight, and $F(\cdot)$ is the Cumulative Distribution Function (CDF). We employ the CDF of noise variables to normalize distances across various noise distributions, ensuring these distances fall within the range of $[0, 1]$, as noise distributions are not identifiable.

**Weights in the Distance**. The noise variables are independent of each other and thus, unlike in perception distance, the change of causal earlier noise variables will not lead to the change of causal later noise nodes. Therefore, if the weight on the difference of each noise variable is the same, the distance will not prefer less change on variables closer to $\mathbf{A}$. To achieve backtracking as little as possible, we set a weight $w_j$ for each node $\mathbf{U}_j$, defined as the number of endogenous decedents of $\mathbf{V}_j$ denoted as $ND(\mathbf{V}_j)$. Generally speaking, for all variables causally earlier than $\mathbf{A}$, one way is to use the number of variables influenced by a particular intervention as the measure of the changes caused by the intervention. Hence, the number of variables influenced by a variable's intervention can be treated as the coefficient of distance. For example, in a causal graph where where $\mathbf{B}$ causes $\mathbf{A}$ and $\mathbf{C}$ is the confounder of $\mathbf{A}$ and $\mathbf{B}$. If $change(\mathbf{A} = \mathbf{a}^*)$, $\mathbf{u_A}$'s and $\mathbf{u_B}$'s weight is 1 and 2 respectively. In this way, variables (e.g., $\mathbf{u_B}$) with bigger influence on other variables possess bigger weights and thus tend to change less, reflecting necessary backtracking.

### G.1  Concretization of Natural Counterfactuals: An Example Methodology

**A Method Based on Mechanism Distance.** We plugin mechanism distance Eqn. 11 into FIO framework Eqn. 2. Below is the equation of optimization:

$$\min_{\mathbf{u}_{an(\mathbf{A})^*}} \sum_{\mathbf{u}_j \in \mathbf{u}_{an(\mathbf{A})}, \mathbf{u}_j^* \in \mathbf{u}_{an(\mathbf{A})^*}} w_j |F(\mathbf{u}_j) - F(\mathbf{u}_j^*)|$$
$$s.t. \quad \mathbf{a}^* = f_A(pa_{\mathbf{A}}^*, \mathbf{u}_{\mathbf{A}}^*) \tag{12}$$
$$s.t. \quad \epsilon < F(\mathbf{u}_j^*) < 1 - \epsilon, \forall \mathbf{u}_j^* \in u_{an(\mathbf{A})^*}$$

Where the first constraint is to achieve $change(\mathbf{A} = \mathbf{a}^*)$, the second constraint require counterfactual data point to satisfy $\epsilon$-natural generation given the optional naturalness criteria (3) in Sec. 4.1, and the

optimization parameter is the value $\mathbf{u}_{an(\mathbf{A})^*}$ of noise variable set $\mathbf{U}_{\mathbf{AN}(\mathbf{A})}$ given $\mathbf{AN}(\mathbf{A}) = an(\mathbf{A})^*$. For simplicity, we use $\mathbf{A}$ as a subscript as an indicator of terms related to $\mathbf{A}$, instead of a number subscript. In practice, the Lagrangian method is used to optimize our objective function. The loss is as below:

$$
\begin{aligned}
\mathcal{L} = \sum_{\mathbf{u}_j \in \mathbf{u}_{an(\mathbf{A})}, \mathbf{u}_j^* \in \mathbf{u}_{an(\mathbf{A})^*}} & w_j |F(\mathbf{u}_j) - F(\mathbf{u}_j^*)| \\
+ w_\epsilon \sum_{\mathbf{u}_j \in \mathbf{u}_{an(\mathbf{A})}, \mathbf{u}_j^* \in \mathbf{u}_{an(\mathbf{A})^*}} & \max((\epsilon - F(\mathbf{u}_j^*), 0) + \max(\epsilon + F(\mathbf{u}_j^*) - 1, 0)) \\
s.t. \quad \mathbf{u}_{\mathbf{A}}^* = f_A^{-1}&(\mathbf{a}^*, pa_{\mathbf{A}}^*)
\end{aligned}
\tag{13}
$$

**In the next section, we use Eqn. 13 for feasible intervention optimization across multiple machine learning case studies, showing that mechanism distance is as effective as perception distance, as discussed in the main paper.**

## G.2 Case Study

### G.2.1 MorphoMNIST

Table 13: MorphoMNIST results of $change(i)$ or $do(i)$ using V-SCM

| Intersection between Ours and NB | | (NC=1, NB=1) | (NC=1, NB=0) | (NC=0, NB=1) | (NC=0, NB=0) |
|---|---|---|---|---|---|
| Number of Intersection | | 5841 | 3064 | 0 | 1094 |
| Non-backtracking | $t$'s MAE | 0.286 | 0.159 | 0.462 | 0.000 | 0.471 |
| | $i$'s MAE | 6.62 | 4.00 | 8.88 | 0.000 | 14.2 |
| Ours | $t$'s MAE | 0.175 | 0.159 | 0.206 | 0.000 | 0.471 |
| | $i$'s MAE | 4.41 | 4.00 | 5.19 | 0.000 | 14.2 |

In this section, we study two types of counterfactuals on the dataset called MorphoMNIST, which contains three variables $(t, i, x)$. From the causal graph shown in Fig. 8 (a), $t$ (the thickness of digit stroke) is the cause of both $i$ (intensity of digit stroke) and $x$ (images) and $i$ is the direct cause of $x$. Fig. 8 (b) shows a sample from the dataset. The dataset contains 60000 images as the training set and 10000 as the test set.

We follow the experimental settings of simulation experiments in Sec. 6.1, except for two differences. One is that we use two state-of-the-art deep learning models, namely V-SCM [25] and H-SCM [28], as the backbones to learn counterfactuals. They use normalizing flows to learn causal relationships among $x$'s parent nodes, e.g., $(t, i)$ in MorphoMNIST. Further, to learn $p(x|t, i)$, notice that V-SCM uses VAE [17] and HVAE [21]. Another difference is that, instead of estimating the outcome with MAE, we follow the same metric called counterfactual effectiveness in [28] developed by [22], First, trained on the dataset, parent predictors given a value of $x$ can predict parent values, i.e., $(t, i)$'s, and then measure the absolute error between parent values after hard intervention or LBF intervention and their predicted values, which is measured on image the pretrained causal model generates given the input of $(t, i)$.

Table 14: Ablation Study on $\epsilon$

| Model | $\epsilon$ | CFs | do($t$) | | do($i$) | |
|---|---|---|---|---|---|---|
| | | | $t$ | $i$ | $t$ | $i$ |
| V-SCM | - | NB | 0.336 | 4.51 | 0.286 | 6.62 |
| | $10^{-4}$ | | 0.314 | 4.48 | 0.197 | 4.90 |
| | $10^{-3}$ | Ours | 0.297 | 4.47 | 0.175 | 4.41 |
| | $10^{-2}$ | | 0.139 | 4.35 | 0.151 | 3.95 |
| H-SCM | - | NB | 0.280 | 2.54 | 0.202 | 3.31 |
| | $10^{-4}$ | | 0.260 | 2.49 | 0.117 | 2.23 |
| | $10^{-3}$ | Ours | 0.245 | 2.44 | 0.103 | 2.03 |
| | $10^{-2}$ | | 0.0939 | 2.34 | 0.0863 | 1.87 |

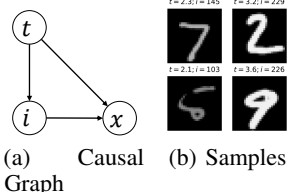

(a)   Causal
Graph     (b) Samples

Figure 8: Causal Graph and samples of MorphoMNIST.

**Quantitative Results of** $change(i)$ **or** $do(i)$**.** We use V-SCM to do counterfactual task of $change(i)$ (where $\epsilon = 10^{-3}$) or $do(i)$ with multiple random seeds on test set. In Table 13, the first column shows the MAE of $(t, i)$, indicating our results outperform that of non-backtracking. Next, we focus on the rest four-column results. In both types of counterfactuals, we use the same value $i$ in $do(i)$ and $change(i)$. Hence, after inference, we know which image satisfies $\epsilon$-natural generation in the two types of counterfactuals. In "NC=1" of the table, NC indicates the set of counterfactuals after feasible intervention optimization. Notice that NC set does not mean the results of natural counterfactuals, since some results do still not satisfy $\epsilon$-natural generation after feasible intervention optimization. "NC=1" mean the set containing data points satisfying $\epsilon$-natural generation and "NC=0" contains data not satisfying $\epsilon$-natural generation after feasible intervention optimization. Similarly, "NB=1" means the set containing data points satisfying naturalness criteria. (NC=1, NB=1) presents the intersection of "NC=1" and "NB=1". Similar logic is adopted to the other three combinations. The number of counterfactual data points is 10000 in two types of counterfactuals.

In (NC=1, NB=1) containing 5841 data points, our performance is similar to the non-backtracking, showing feasible intervention optimization tends to backtrack as less as possible when hard interventions have satisfied $\epsilon$-natural generation. In (NC=1, NB=0), there are 3064 data points, which are "unnatural" points in non-backtracking counterfactuals. After natural counterfactual optimization, this huge amount of data points becomes "natural". In this set, our approach contributes to the maximal improvement compared to the other three sets in Table 13, improving 55.4% and 41.6% on thickness $t$ and intensity $i$. The number of points in (NC=0, NB=1) is zero, showing the stability of our algorithm since our approach will not move the hard, feasible intervention into an unfeasible intervention. Two types of counterfactuals perform similarly in the set (NC=0, NB=0), also showing the stability of our approach.

**Ablation Study on Naturalness Threshold** $\epsilon$**.** We use two models, V-SCM and H-SCM, to do counterfactuals with different values of $\epsilon$. As shown in Table 14, our error is reduced as the $\epsilon$ increases using the same inference model, since the higher $\epsilon$ will select more feasible interventions.

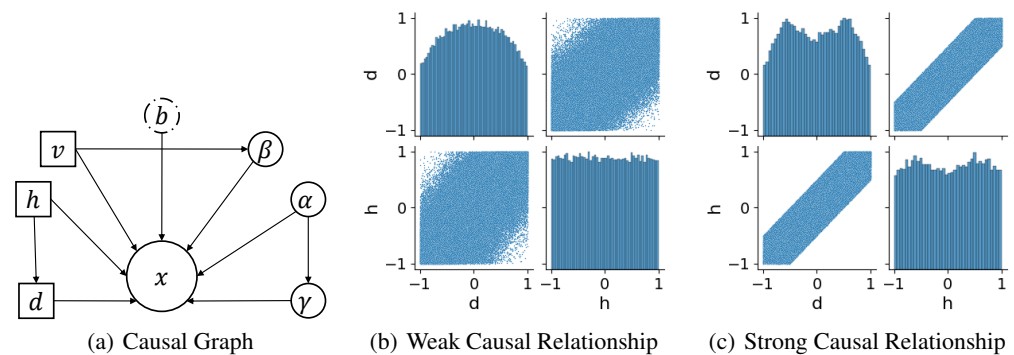

(a) Causal Graph     (b) Weak Causal Relationship     (c) Strong Causal Relationship

Figure 9: Causal graph of 3DIdent and the causal relationships of variables $(d, h)$ in Weak-3DIdent and Strong-3DIdent respectively.

### G.2.2  3DIdentBOX

In this task, we utilize practical public datasets called 3DIdentBOX, which encompass multiple datasets [4]. Specifically, we employ Weak-3DIdent and Strong-3DIdent, both of which share the

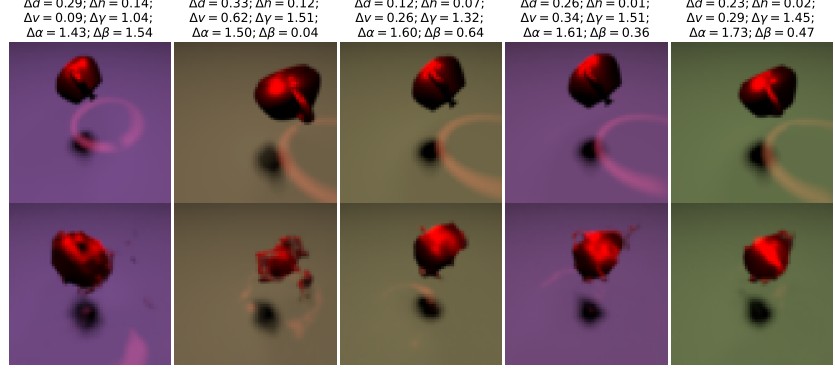

Δd = 0.29; Δh = 0.14; Δv = 0.09; Δγ = 1.04; Δα = 1.43; Δβ = 1.54  | Δd = 0.33; Δh = 0.12; Δv = 0.62; Δγ = 1.51; Δα = 1.50; Δβ = 0.04 | Δd = 0.12; Δh = 0.07; Δv = 0.26; Δγ = 1.32; Δα = 1.60; Δβ = 0.64 | Δd = 0.26; Δh = 0.01; Δv = 0.34; Δγ = 1.51; Δα = 1.61; Δβ = 0.36 | Δd = 0.23; Δh = 0.02; Δv = 0.29; Δγ = 1.45; Δα = 1.73; Δβ = 0.47

(a) Results of Non-backtracking Counterfactuals

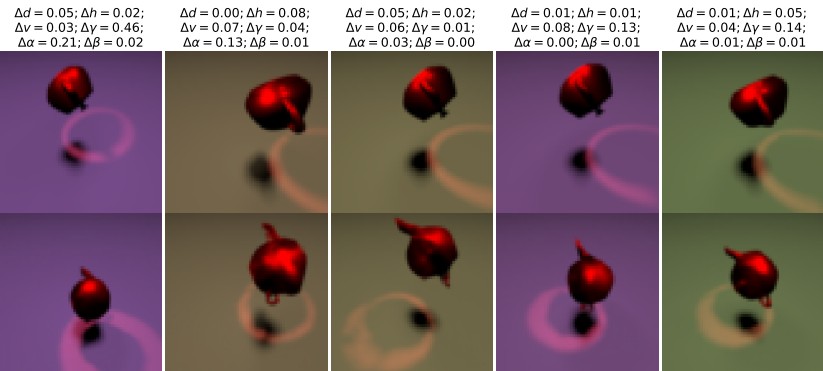

Δd = 0.05; Δh = 0.02; Δv = 0.03; Δγ = 0.46; Δα = 0.21; Δβ = 0.02 | Δd = 0.00; Δh = 0.08; Δv = 0.07; Δγ = 0.04; Δα = 0.13; Δβ = 0.01 | Δd = 0.05; Δh = 0.02; Δv = 0.06; Δγ = 0.01; Δα = 0.03; Δβ = 0.00 | Δd = 0.01; Δh = 0.01; Δv = 0.08; Δγ = 0.13; Δα = 0.00; Δβ = 0.01 | Δd = 0.01; Δh = 0.05; Δv = 0.04; Δγ = 0.14; Δα = 0.01; Δβ = 0.01

(b) Results of Natural Counterfactuals

Figure 10: Visualization Results on Stong-3DIdent.

Table 15: Results on Weak-3DIdent and Stong-3DIdent

| Dataset | Counterfactuals | $d$ | $h$ | $v$ | $\gamma$ | $\alpha$ | $\beta$ | $b$ |
|---|---|---|---|---|---|---|---|---|
| Weak-3DIdent | Non-backtracking | 0.0252 | 0.0191 | 0.0346 | 0.364 | 0.266 | 0.0805 | 0.00417 |
|  | Ours | 0.0241 | 0.0182 | 0.0339 | 0.348 | 0.224 | 0.0371 | 0.00416 |
| Stong-3DIdent | Non-backtracking | 0.104 | 0.0840 | 0.0770 | 0.385 | 0.495 | 0.338 | 0.00476 |
|  | Ours | 0.0633 | 0.0512 | 0.0518 | 0.326 | 0.348 | 0.151 | 0.00464 |

same causal graph depicted in Fig. 9 (b), consisting of an image variable denoted as $x$ and seven parent variables. These parent variables, denoted as $(d, h, v)$, control the depth, horizon position, and vertical position of the teapot in image $x$ respectively. Additionally, the variables $(\gamma, \alpha, \beta)$ govern three types of angles associated with the teapot within images, while variable $b$ represents the background color of the image. As illustrated in Fig. 9 (a), causal relationships exist among three pairs of parent variables, i.e., $(h, d)$, $(v, \beta)$ and $(\alpha, \gamma)$. It is important to note a distinction between Weak-3DIdent and Strong-3DIdent. In Weak-3DIdent, there exists a weak causal relationship between the variables of each pair, as shown in Fig. 9 (b), whereas in Strong-3DIdent, the causal relationship is stronger, as depicted in Fig. 9 (c).

We follow the same experimental setup as in the MophoMNIST experiments. Using an epsilon value of $\epsilon = 10^{-3}$ we employ the H-SCM as the inference model. We conduct interventions or changes on the variables $(d, \beta, \gamma)$ and the results are presented in Table 15. In both datasets, our approach outperforms the non-backtracking method, with Strong-3DIdent exhibiting a more significant margin over the non-backtracking method. This is because the non-backtracking method encounters more unfeasible interventions when performing hard interventions using Strong-3DIdent.

Additionally, we perform visualizations on Strong-3DIdent. In Fig. 10, we display counterfactual outcomes in (a) and (b), where the text above each image in the first row (evidence) indicates the error in the corresponding counterfactual outcome shown in the second row. In Fig. 10 (a), we present counterfactual images that do not meet the $\epsilon$-natural generation criteria in the non-backtracking approach. In contrast, Fig. 10 (b) showcases our results, which are notably more visually effective. This demonstrates that our solution can alleviate the challenges posed by hard interventions in the non-backtracking method.

