# OpenReview forum: "Natural Counterfactuals With Necessary Backtracking"
_NeurIPS.cc/2024/Conference — NeurIPS 2024 poster_

### Official Review · Reviewer_fYWn · 2024-07-02

**Soundness:** 3
**Presentation:** 3
**Contribution:** 4
**Rating:** 7
**Confidence:** 3

**Summary:**

The authors propose a novel approach for generating causally valid and *natural* counterfactuals. These counterfactuals are *natural* in that they are close to the observed data manifold. This is achieved by allowing a certain amount of backtracking, which involves tracing back upstream causal effects on variables.

The authors essentially propose a more flexible version of Judea Pearl’s $do$-calculus, by replacing the explicitly non-backtracking $do$-operator with a $change$-operators that represents the notion of a *least-backtracking feasible intervention* (LBF). Intuitively, LBF interventions take into account the effects of causal ancestors instead of considering the intervention itself as entirely exogenous.

In order to compute LBF interventions, the authors propose a *feasible intervention optimization* that targets minimal changes to the causal ancestors of some variable $A$ while satisfying certain naturalness constraints. These constraints ensure that the generated counterfactual remains anchored in the observed data manifold. Through experiments involving both synthetic and real-world datasets, the authors demonstrate that their proposed method yields more “natural” counterfactuals than the non-backtracking alternative approach.

The proposed method has some clear links to the literature on counterfactual explanations, which are mentioned but not thoroughly explored here. This is an interesting avenue for future research and this work constitutes a solid first step in that direction. Overall, the paper seems to make a strong contribution to the field of Causal Inference, although I am not a subject expert.

**Strengths:**

- Interesting and natural extension of non-backtracking counterfactual generation.
- Empirical results for both synthetic and real-world datasets show improved performance when LFB is used.

**Weaknesses:**

- No standard errors reported in Tables, so difficult to assess how substantial the performance differences are after accounting for noise.
- Link to CE is mentioned, but the authors could have gone into more detail. In particular, you propose that the literature on CE could benefit from incorporating your definition of *naturalness* (which I don’t disagree with) but you fall short of comparing it to existing definitions in CE. See also my related questions.
- The figures and annotations are too small (and in general, both figures and tables currently make the paper look a little crammed).

**Questions:**

### Link to CE

- How does this approach compare to existing approaches to causal algorithmic recourse (e.g. [MINT](https://arxiv.org/abs/2002.06278))?
- How do the proposed *naturalness* constraints compare to *plausibility* (e.g.  [Artelt and Hammer](https://arxiv.org/abs/2002.04862), [REVISE](https://arxiv.org/abs/1907.09615)) and *feasibility* (e.g. [FACE](https://arxiv.org/abs/1909.09369)) constraints in CE/algorithmic recourse? Equation 2 in your paper looks quite similar to Equation 5 in [Artelt and Hammer](https://arxiv.org/abs/2002.04862).

### Other questions

- The sentence spanning from line 81 to 86 is very long and it’s easy to get lost in it. Consider splitting this into 2/3 sentences.
- Figures and annotations on page 8 are very small (annotations are barely legible).
- Could you highlight the single point from Figure 1 (a) also in panel (b)? To illustrate that in (b) you essentially repeat the experiment over all test samples.

**Limitations:**

The authors mention that the assumption of invertibility is a small limitation, but not theoretically problematic. The work may have other limitations that I have failed to recognized, as I’m not a subject expert.

---

> ### Author Rebuttal · Authors · 2024-08-07
>
> ***Weaknesses***
>
> **1. No standard errors reported in Tables, so difficult to assess how substantial the performance differences are after accounting for noise.**
>
> Thank you for asking this. We have included the standard errors of all experiments in Tables 9-12 in Section A.4 of the Appendix. As you can see, the standard deviation for our natural counterfactuals is generally lower.
>
>
> **2. The figures and annotations are too small (and in general, both figures and tables currently make the paper look a little crammed).**
>
>
> Thank you. We have made modifications according to your helpful suggestions.
>
> ***Questions Linked to CE***
>
> **1. How does this approach compare to existing approaches to causal algorithmic recourse (e.g. MINT)?**
>
> Thank you for this great question. We would like to mention that the proposed work has different motivations and purposes. Our natural counterfactuals aim to provide a principled inference method to ensure that counterfactual scenarios remain realistic relative to the regular causal mechanisms,  with the actual (conditional) data distribution (using a limited extent of backtracking) as a surrogate of the mechanisms. Given an expected change or intervention, we offer a framework for inferring counterfactual outcomes. In contrast, causal algorithmic recourse focuses on identifying interventions to achieve a pre-specified desirable counterfactual outcome. In summary, similar to Pearl's counterfactuals, natural counterfactuals aim to perform counterfactual inference to obtain counterfactual outcomes, while causal algorithmic recourse seeks interventions to achieve a given desirable outcome.
>
> However, if we assume the aim is to find intervention variables to achieve $A = a^*$ in our framework, the task becomes more similar to finding interventions that cause a given outcome, as in causal algorithmic recourse. Still, our framework does not operate in exactly this way. Instead, we identify a group of intervention variables $C$ that includes $A$, which does not imply that intervening on variables other than $C$ causes $A$. This means that we perform interventions on all variables in $C$, including $A$, simultaneously. In other words, our aim is to find, so to speak, companion interventions in order to make the result ``natural'' in our sense.
>
>
> **2. How do the proposed naturalness constraints compare to plausibility (e.g. Artelt and Hammer, REVISE) and feasibility (e.g. FACE) constraints in CE/algorithmic recourse? Equation 2 in your paper looks quite similar to Equation 5 in Artelt and Hammer.**
>
>
> Thank you for recommending these useful works. We have discussed them in our updated manuscript, including the paper mentioned in Question 1, summarized below.
>
> 1. Our proposed naturalness constraints share a similar spirit with the feasibility constraints in counterfactual explanation (CE)/algorithmic recourse, as existing methods may overlook real-world constraints. However, there are two key differences:
>
>    - **Different Motivations:** We propose naturalness constraints for counterfactual inference, which generates counterfactual outcomes that are natural with respect to the actual world. In contrast, the feasibility constraints of CE/algorithmic recourse serve to identify possible interventions that can lead to a given desired outcome.
>
>    - **Scope of Application:** While feasibility constraints may consider more detailed constraints for a specific application, we propose a general framework without considering the meanings of specific variables. Of course, when applying our framework in a specific situation, users can and should consider special constraints relevant to that situation.
>
> 2. The reason why Equation 2 in our paper looks similar to Equation 5 in [Artelt and Hammer](https://arxiv.org/pdf/2002.04862) is that both involve optimization with constraints related to the observed distribution. However, Artelt and Hammer focus on CE, and the two papers are fundamentally different:
>
>    - **Different Purposes:** CE is similar to algorithmic recourse in that it seeks to find the input for a target output. Our ultimate aim is to obtain a feasible output given an input.
>
>    - **Framework Differences:** Unlike algorithmic recourse, CE does not build on the SCM framework, making it difficult to explain the similarity between actual and counterfactual values in CE. Additionally, CE often studies anti-causal problems, such as classifying an image $x$ into a category $y$, where $y$ is the cause of $x$ instead of being the causal outcome. Although CE tries to minimize the difference between actual and counterfactual values, it is hard to claim that the distance is minimized from a causal perspective.
>
>
>
> ***Other questions***
>
> **The sentence spanning from line 81 to 86 is very long and it’s easy to get lost in it. Consider splitting this into 2/3 sentences.** and **Figures and annotations on page 8 are very small (annotations are barely legible).**
>
> Thank you for the feedback. We have implemented your helpful suggestions to modify lines 81-86 and updated the figures and annotations on page 8.
>
> **Could you highlight the single point from Figure 1 (a) also in panel (b)? To illustrate that in (b) you essentially repeat the experiment over all test samples.**
>
> Thank you for your excellent suggestion. In addition to your suggestions, we have sampled two more data points. Consequently, we have displayed both the non-backtracking counterfactual values and our natural counterfactual values for three data points. Specifically, we randomly select three data points that require backtracking in our natural counterfactuals. Moreover, when a hard intervention is feasible, our natural counterfactuals and non-backtracking counterfactuals yield the same results, so we do not sample these cases. The updated PDF file with the related figures has been uploaded under the global response.

---

> > ### Comment · Reviewer_fYWn · 2024-08-09
> > **Response to rebuttal**
> >
> > Thanks for the detailed rebuttal, I appreciate the effort and have no further questions at this point

---

### Official Review · Reviewer_JBox · 2024-07-07

**Soundness:** 2
**Presentation:** 3
**Contribution:** 2
**Rating:** 5
**Confidence:** 4

**Summary:**

The paper presents a framework for generating "natural counterfactuals" that are more feasible within the support of  the training data distribution. This approach includes controlled backtracking through an optimization method  that uses a "naturalness" criterion as a constraint.

**Strengths:**

Original combination of existing ideas, e.g., Pearl's non-backtracking ( interventional ) counterfactuals, backtracking (observational) counterfactuals, normalizing flows to learn causal mechanisms

**Weaknesses:**

My first concern is with the term "natural counterfactual." Both Pearl's non-backtracking counterfactuals and backtracking counterfactuals use an algorithm that begins with the prior probabilities of the exogenous variables. In the abduction step, the counterfactual posterior is computed based on the observed facts in the non-backtracking case, or on both the observed facts and observed counterfacts in the backtracking case. However, this paper samples from the observational distribution of the training data.

Theorem 1 builds on prior work $[17]$ where the posterior distribution has been discussed, mentioning that due to the monotonicity assumption of $f_i$ with respect to
$U$, the posterior distribution is a point mass on $𝑈=u$. However, the experiments in this paper sample from the prior distributions of the structural causal model (SCM), which means it is not a true counterfactual distribution.

This leads me to the purpose of the paper. It uses optimization to find a point within a high-density area of the data manifold generated by the observational distribution, given a request to change a value. Which use case would require this result?
For example, Pearl’s non-backtracking counterfactuals address questions on actual cause, causal necessity, and personalized policy. If performing $do(A)$ is not feasible, it does not imply that a feasible data point within the training data distribution can answer these questions. Further, the concept of "naturalness" is not based on the immutability of the variable but on the probability density of the variable given its parents. For example, the variable "Age" might meet the criterion based on its probability density, but it remains immutable.

References:

[17] Chaochao Lu, Biwei Huang, Ke Wang, José Miguel Hernández-Lobato, Kun Zhang, and
      Bernhard Schölkopf. Sample-efficient reinforcement learning via counterfactual-based data
     augmentation. arXiv preprint arXiv:2012.09092, 2020.

**Questions:**

(1) Line 46: “When interventions lead to unrealistic scenarios relative to the training data,  predicting counterfactual outcomes in such scenarios can be highly uncertain and inaccurate [12].”
Reference [12] addresses selection bias in potential outcome counterfactuals, in particular,  when the treatment group does not match the control group. It is not related to SCM counterfactuals.

(2) Line 48 “This issue becomes particularly pronounced when non-parametric models are employed, as they  often struggle to generalize to unseen, out-of-distribution data [27]. “
Reference [27] discusses the issue of i.i.d. in machine learning, particularly when the training data distribution differs from the interventional distribution, leading to out-of-distribution scenarios. How is the current paper related  to the discussion in [27]?

(3) Appendix F: “Therefore, [30]’s backtracking counterfactual does not reduce to Pearl’s counterfactual even when $A^∗ =\emptyset$.”
This is incorrect. Neither non-backtracking nor backtracking is a special case of the other. Appendix A of [30] presents a unified framework for counterfactual reasoning that integrates both backtracking and non-backtracking counterfactuals, as suggested by an area chair at that conference. Also, if $A^∗ =\emptyset$, then there is no counterfactual query.

References:

[12] Negar Hassanpour and Russell Greiner. Learning disentangled representations for counterfactual
      regression. In International Conference on Learning Representations, 2019.

[27] Bernhard Schölkopf, Francesco Locatello, Stefan Bauer, Nan Rosemary Ke, Nal Kalchbrenner,
      Anirudh Goyal, and Yoshua Bengio. Toward causal representation learning. Proceedings of the
     IEEE, 109(5):612–634, 2021

[30] Julius von Kügelgen, Abdirisak Mohamed, and Sander Beckers. Backtracking counterfactuals.
      arXiv preprint arXiv:2211.00472, 2022.

---

> ### Author Rebuttal · Authors · 2024-08-07
>
> ***Weaknesses***
>
>
> **(1) My first concern is ... However, the experiments in this paper sample from the prior distributions of ... counterfactual distribution.**
>
> Thanks for raising the concern. To clarify, we sample from the posterior distribution of $U$. First, due to the monotonicity assumption, sampling from the observational distribution amounts to sampling from the distribution of the exogenous variables. More importantly, during inference, the only difference is that non-backtracking counterfactuals use interventions on $A$, while natural counterfactuals use interventions on $C$, with observed evidence considered in both. Hence, we are sampling from posterior distribution of $U$. We have made this clearer in updated manuscript.
>
> Your further feedback would be appreciated.
>
> **(2) This leads me to the purpose of the paper. ... answer these questions.**
>
> This is a good point that we had in mind before starting the project. We have revised the presentation to highlight our motivation and explain why we prefer feasible interventions for addressing counterfactual queries. We are motivated by observing that non-backtracking counterfactual generation, for all its merits, often results in scenarios that are not realistic (and so, as a consequence, unreliable when it is based on training data). On the other hand, we also think that fully backtracking counterfactuals do not have much implication for guiding actions, because they appeal to intervention on unobserved or even unobservable, unidentifiable variables. We thus develop a framework that generates realistic counterfactuals by finding feasible interventions on observed variables while minimizing backtracking. In particular, when the intervention set $C$ picked out by our algorithm is sufficiently simple (a special case is that it is identical to original target set $A$), the counterfactual will remain useful for guiding actions.
>
> As demonstrated by our experiments, generating a non-backtracking counterfactual could be very unreliable, when it is not "natural" in our sense. So one use of our framework is to check on the reliability or feasibility of non-backtracking counterfactuals, and when dubious, to generate a reliable surrogate by adjusting the antecedent of the counterfactual. We think that reliable information about the adjusted counterfactuals will be more useful than probably unreliable information about the original, non-backtracking counterfactuals.
>
> **(3) Further, the concept of "naturalness" ... but it remains immutable.**
>
> Thanks for raising the issue. As noted in the footnote on page 4, "naturalness" can have various interpretations. In this paper, we provide a general framework for natural counterfactuals without considering the meanings of specific variables. Hence, we do not discuss all possible constraints for various real-world scenarios that deserve to be labeled "naturalness". It is possible and in some situations important to incorporate more constraints, such as the relative ease of intervention, into this framework, though we did not attempt that in this paper. We will consider using another term if "naturalness" is perceived as inapt or misleading. We have included a remark on your good point in updated paper.
>
> ***Questions***
>
> **(1) Line 46: Reference [12] addresses ... ... It is not related to SCM counterfactuals.**
>
>
> Yes, [12] is based on the potential outcome framework, but we believe that one of its points is aligned with our statement: "The fact that counterfactual outcomes are unobservable (i.e., not present in any training data) makes estimating treatment effects more difficult than the generalization problem in the supervised learning paradigm." Still, we are reconsidering whether this reference is more helpful than misleading in our context. Thanks for this helpful comment.
>
> **(2) Line 48: How is the current paper related to the discussion in [27]?**
>
> Thanks for the question. One of our concerns about non-backtracking counterfactuals, despite their novelty and elegance, is precisely their frequent requirement to do out-of-sample generalization. This is why we believe the discussion in [27] of the difficulty of the out-of-distribution problem is very relevant. If you do not think this is convincing to you, please let us know.
>
> **(3) Appendix F: “[30]’s backtracking counterfactual does not reduce to Pearl’s ... no counterfactual query.**
>
> This statement of ours is a bit confusing. Thanks for pointing it out! We did not mean to claim that backtracking counterfactuals in [30] are or should be a special case of Pearl’s counterfactuals. What we meant was that $A^* = \emptyset$ represents a special though extreme case of a counterfactual query, and intuitively, when $A^* = \emptyset$, the counterfactual instance should be exactly the same as the actual instance, for nothing is supposed in the antecedent of the counterfactual. Pearl's counterfactuals agree with this intuition. However, [30]'s backtracking counterfactual does not reduce to this result in this special case, as it allows changes in the exogenous variables even in the case of a null antecedent. Indeed, it would in the special case allow any value setting of the exogenous variables, for they are all compatible with the null antecedent.
>
> Note that another extreme case could illustrate that backtracking counterfactuals proposed in [30] seem to invoke gratuitous changes sometimes. In general, when the counterfactual value $a^*$ is supposed to be equal to the actual value $a$, Pearl's theory will return the exact same instance as the actual one, which is intuitively reasonable, but the theory in [30] will give positive probability to other instances as well. One may stipulate that $a^*$ must be different from $a$ in order to count as a counterfactual query, but to our knowledge, the literature usually understands counterfactuals as just subjunctive conditionals, which also includes subjunctive conditionals with true antecedents as special cases.

---

> > ### Comment · Reviewer_JBox · 2024-08-08
> >
> > Thank you for the responses; I appreciate your efforts.
> >
> > Here is my response:
> >
> > (1) Let's consider the experiment involving the toy 1 example:The prior distribution of the exogenous variables is assumed to be a standard Gaussian. The factual observation is given as $(n_1, n_2, n_3) = (−0.59, 0.71, −0.37)$. Due to the monotonicity of the functions with respect to U, I calculated the values ($u_1, u_2, u_3) = (-0.59, 0.36, -0.93)$. As a result, the posterior counterfactual distribution becomes a point mass (Dirac delta) centered at ($u_1, u_2, u_3) = (-0.59, 0.36, -0.93)$. This means that sampling from this posterior distribution will always gives us the same values, $(u_1, u_2, u_3) = (-0.59, 0.36, -0.93)$. However, if I understand correctly, you are sampling 10,000 times from the prior distribution of the exogenous variables, which is the standard Gaussian.
> >
> > Counterfactual reasoning, whether non-backtracking or backtracking, generally requires knowledge of the Structural Causal Model (SCM). In the absence of an SCM, certain estimates can be made under specific conditions if both observational and interventional data are available, as discussed in a paper by Jin Tian and Judea Pearl. I did not observe any application of non-backtracking or backtracking counterfactual reasoning in this paper. However, the term "counterfactual" has been used in other contexts, similar to this paper, where an optimization method finds the closest point to a given data point. This approach is also seen in works like those by Wachter et al. and subsequent related papers.
> >
> > (2) I still don't see a clear use case or problem statement that can be effectively addressed using the results of your optimization framework. Even in the Toy 1 example, if we make the request $do(n_2=0.19)$, our interest lies in the interventional distribution. This request is necessarily out-of-distribution. How does finding values of $n_1$ and $n_2$ that satisfy the optimization actually solve the problem at hand? It's also worth noting that this approach isn't performing $do(n_1)$ or $do(n_2) $ interventions but rather finding the closest points within the observational distribution.
> >
> > (3) I agree that there could be different different interpretation of "Naturalness". Thanks!
> >
> > Questions:
> >
> > (1) Thanks for the explanation.
> >
> > (2) You wrote: "One of our concerns about non-backtracking counterfactuals, despite their novelty and elegance, is precisely their frequent requirement to do out-of-sample generalization."
> >
> > Out-of-sample generalization is not a concern of non-backtracking (interventional) counterfactuals; instead, it is a necessary feature of it.
> >
> > (3) I read [30] again. They have Property 1 (preference for closeness). In other words, the closest (most similar) world to an actual world is itself. Please note that this is not important for the main issues of the paper.

---

> ### Author Response · Authors · 2024-08-09
> **Response 1-1**
>
> We are truly grateful for your prompt feedback and for providing more details regarding your original concerns. We also appreciate your acknowledgment that "naturalness" can have various interpretations and that you liked the explanation for Reference [12]. We are pleased to address your remaining questions and look forward to your further feedback.
>
> **W(1)-1: Let's consider the experiment involving the toy 1 example: ... $(n_1, n_2, n_3)=(-0.59, 0.71, -0.37)$. Due to the monotonicity of the functions with respect to U, I calculated the values $(u_1, u_2, u_3)=(-0.59, 0.36, -0.93)$. As a result, the posterior counterfactual distribution becomes a point mass (Dirac delta) centered at $(u_1, u_2, u_3)=(-0.59, 0.36, -0.93)$. This means that sampling from this posterior distribution will always gives us the same values, $(u_1, u_2, u_3)=(-0.59, 0.36, -0.93)$. However, if I understand correctly, you are sampling 10,000 times from the prior distribution of the exogenous variables, which is the standard Gaussian.**
>
> Thank you for this question, and please see our response below.
>
> First, *our paper does not mention sampling 10,000 times from the prior distribution of the exogenous variables.* Instead, our test set comprises 10,000 data points, and we form 10,000 counterfactual queries to assess the effectiveness of counterfactual inference. Specifically, for each query, we use one data point as evidence, such as $(n_1, n_2, n_3) = (-0.59, 0.71, -0.37)$, and randomly sample a value for $n_2$ from its distribution as the counterfactual supposition, such as $n_2 = 0.19$.
>
> Second, for any query, our inference procedure can be seen as following Pearl's three-step process: (1) Update the noise distribution given the evidence to a posterior; (2) Modify the SCM using the identified "natural intervention"; (3) Conduct inference using the posterior noise distribution and the modified SCM. As stated in Line 146, the only difference from the non-backtracking inference lies in Step (2), where we perform a least-backtracking feasible intervention on the variable set $C$, which consists of some causal ancestors of the variable set $A$ in addition to $A$.  Notably, when the intervention on $A$ alone already satisfies our criteria, we have $C = A$, yielding the non-backtracking counterfactual.
>
> Specifically, in our method as well as in Pearl's, *the first step* involves updating the prior exogenous distribution $p(U)$ to the posterior distribution given the evidence, and in the example you mentioned, would yield a point mass distribution on $U=(-0.59, 0.36, -0.93)$, as you noted. In our implementation, normalized flows ensure that by inputting $(n_1, n_2, n_3) = (-0.59, 0.71, -0.37)$ into our model, we obtain the unique posterior value of $U$ (as functions are invertible).
>
> *The second step* involves modifying the SCM. In non-backtracking counterfactuals, an intervention $do(n_2 = 0.19)$ is applied, without changing anything in $n_2$'s causal upstream. In contrast, natural counterfactuals may invoke some backtracking if needed, and in this case would apply a least-backtracking feasible intervention $do(n_1 = -0.02, n_2 = 0.19)$ obtained through our Feasible Intervention Optimization, which would yield an instance that satisfies our "naturalness" criterion.
>
> *In the final step*, both methods utilize the updated noise distribution (now a single point) and the modified SCM to perform inference and obtain the outcome value of $n_3$.
>
> We have refined the presentation of our inference procedure in the updated manuscript. Thanks again for giving us the opportunity to clear misunderstanding.
>
> **W(1)-2: In the absence of an SCM, certain estimates can be made under specific conditions if both observational and interventional data are available, as discussed in a paper by Jin Tian and Judea Pearl. I did not observe any application of non-backtracking or backtracking counterfactual reasoning in this paper.**
>
> Sorry, we are not sure what you meant by "application of non-backtracking or backtracking counterfactual reasoning"--your further feedback would be helpful. In this paper, we made some simplifying assumptions, compared to the classical work on (non-backtracking) counterfactual inference by Pearl and his collaborators (Tian, Shpister, Bareinboim, etc.), especially the assumption of no latent confounding (our causal diagram is a DAG with no bi-directed arcs), along with the assumptions of access only to observational distribution and monotonicity. As a result, we need not invoke the full machinery of those seminal works. Still, as explained above, we are effectively following Pearl's three-step procedure.
>
> Perhaps we misunderstood your concern here and, if so, would very much appreciate your elaboration.

---

> ### Author Response · Authors · 2024-08-09
> **Response 1-2**
>
> **W(1)-3: However, the term "counterfactual" has been used in other contexts, similar to this paper, where an optimization method finds the closest point to a given data point. This approach is also seen in works like those by Wachter et al. and subsequent related papers.**
>
> Thank you for the opportunity to clarify the differences between our paper and Wachter et al's work and much of the literature on counterfactual explanation. We have cited the work of Wachter et al. and related papers, noting that our paper and their work on counterfactual explanations fundamentally differ in purpose, techniques, and cognitive frameworks.
>
> The term "counterfactual" in such counterfactual explanations is unrelated to "counterfactual" in causal inference, as they do not rely on causality or the SCM framework. For more details, please refer to Section III-C in [Wachter et al.](https://arxiv.org/pdf/1711.00399), which explains that counterfactual explanations do not employ causal assumptions. Typically, in counterfactual explanations, given an input (image) $x$ and an output (class) $y$ from a model, the goal is to find another input $x'$ that leads to a different output $y'$ while ensuring $x'$ is as similar as possible to the original input $x$ based on their value difference. Although counterfactual explanations are unrelated to causality, they borrow causal terminology, such as "counterfactual," to refer to $x'$. This helps explain the classification behavior of the given model.
>
> We can summarize the key differences between our paper and counterfactual explanations as follows:
>
> - **Different Purposes and Cognitive Frameworks**: Based on a general machine learning framework, counterfactual explanations aim to explain model behavior by finding another input with minimal changes that results in a different class. In contrast, similar to Pearl's counterfactuals, natural counterfactuals perform counterfactual inference to obtain counterfactual outcomes based on a causal framework, where we also consider the feasibility of interventions and prefer the least backtracking.
>
> - **Different Technologies**: Abstractly, our method infers an outcome given an input, while counterfactual explanations find an input given a target output.
>
> - **Differences from a Causal Perspective**: Although Wachter et al.'s counterfactual explanations are unrelated to causality, from a causal perspective, they often address anti-causal problems, such as classifying an image $x$ into a category $y$, where $y$ is treated as the cause of $x$, as commonly seen in causal inference literature. Since counterfactual explanations do not build on the SCM framework, even though they try to minimize the value difference between $x'$ and $x$, it is difficult to interpret the distance from a causal perspective, unlike in our approach, which is explicitly based on causal assumptions as given in a causal DAG.

---

> ### Author Response · Authors · 2024-08-09
> **Response 1-3**
>
> **W(2)-1: I still don't see a clear use case or problem statement that can be effectively addressed using the results of your optimization framework. Even in the Toy 1 example, if we make the request $do(n_2=0.19)$, our interest lies in the interventional distribution. This request is necessarily out-of-distribution. How does finding values of $n_1$ and $n_2$ that satisfy the optimization actually solve the problem at hand?**
>
> Thank you for elaborating on your concern. A general counterfactual query takes the following form: Given evidence $E$, what would the value of $B$ have been if $A$ had taken the value $a^*$? Non-backtracking counterfactuals, fully backtracking counterfactuals, and natural counterfactuals offer three different interpretations of this general query.
>
> Specifically, in non-backtracking counterfactuals, the counterfactual supposition $A = a^*$ is interpreted to be realized by directly intervening on $A$ alone, i.e., using $do(A)$, as you mentioned. In fully backtracking counterfactuals, [30] interprets the counterfactual supposition to be realized by changing (which amounts to intervening on) exogenous variables. In our natural counterfactuals, we in general realize the counterfactual supposition by intervening on $C$, which includes $A$ and possibly some of $A$'s causal antecedents, to ensure that the intervention is feasible in our sense. As we can see, the three methods use different interventions about the supposition in a counterfactual query, and neither fully backtracking counterfactuals nor our natural counterfactuals interpret a counterfactual query as necessarily a query about the distribution resulting from intervening on $A$ alone.
>
> In our framework of natural counterfactuals, we aim to generate counterfactuals that are "natural" with respect to the actual distribution but otherwise stay as close to non-backtracking as possible. Therefore, we use Feasible Intervention Optimization to find the least-backtracking feasible intervention on $C$ within the support of the actual distribution by minimizing the extent of backtracking, as explained in Question W(1)-1. For example, in Question W(1)-1, the set $C = (n_1, n_2)$, obtained through our optimization, is used to answer the query.
>
> Additionally, similar to our natural counterfactuals, backtracking counterfactuals in [30] essentially use data points from the observed distribution to answer $A = a^*$ as well. The difference is that we intervene on endogenous variables, which is potentially still useful for guiding actions, while [30] intervenes on exogenous variables, which is not useful for guiding actions, as far as we can see.
>
> You wrote: *"Even in the Toy 1 example, if we make the request $do(n_2 = 0.19)$, our interest lies in the interventional distribution. This request is necessarily out-of-distribution."*  In the sense of "out-of-distribution" used in our paper, we respectfully think this is not true. An intervention, though cutting out some mechanisms originally in place, does NOT necessarily result in an instance outside the support of the observed distribution. For example, although the request $do(n_2 = 0.19)$ in the Toy 1 example is indeed out-of-distribution in our sense, not every intervention is out-of-distribution. For example, given the evidence $(n_1, n_2, n_3) = (-0.59, 0.71, -0.37)$, $do(n_2 = 0.50)$ is feasible, resulting in the counterfactual $(n_1 = -0.59, n_2 = 0.50)$ being within the support of the observable distribution. In our natural counterfactuals, we actually distinguish between in-distribution $do(A = a^*)$ and out-of-distribution $do(A = a^*)$ and adjust out-of-distribution cases by performing a least-backtracking feasible intervention on $A$'s ancestors $C$.

---

> ### Author Response · Authors · 2024-08-09
> **Response 1-4**
>
> **W(2)-2: It's also worth noting that this approach isn't performing or interventions but rather finding the closest points within the observational distribution.**
>
> As explained in Questions W(1)-1 and W(2)-1, our method also performs interventions, following the same three-step inference procedure as non-backtracking counterfactuals: updating the posterior distribution, modifying the SCM by intervention, and conducting inference on the modified SCM. The key difference is that we perform a least-backtracking feasible intervention on set $C$, which usually includes some of $A$'s causal ancestors in addition to $A$. If an intervention on $A$ alone is feasible, then $C = A$.
>
> In the example from W(1)-1, we perform interventions on both $n_1$ and $n_2$ simultaneously, whereas non-backtracking counterfactuals perform an intervention only on $n_2$. Specifically, in natural counterfactuals, both $n_1$ and $n_2$ have their links to their respective parents severed. Since $n_1$'s only parent node is $u_1$, the link between $n_1$ and $u_1$ is disconnected, and $n_1$ is directly assigned the value $-0.02$. Similarly, $n_2$'s links to $u_2$ and $n_1$ are severed, and $n_2$ is set to $0.19$.
>
> As noted in Question W(2)-1, when intervention $do(A)$ alone is feasible, i.e., $C = A$, our counterfactual outcome matches the non-backtracking outcome. For example, given the evidence $(n_1, n_2, n_3) = (-0.59, 0.71, -0.37)$, $do(n_2 = 0.50)$ is feasible, meaning that $(n_1 = -0.59, n_2 = 0.50)$ falls within the support of the observable distribution. In this instance, the inference is the same for both natural and non-backtracking counterfactuals. During the second step of inference, $n_2$'s links to $u_2$ and $n_1$ are severed, and $n_2$ is set to $0.50$.
>
> To reiterate, our approach is a *causal* approach, relying on a given causal structure represented by a DAG. It is not simply minimizing a non-causal distance within the observational distribution.
>
> **Questions:**
>
> **Q(2): You wrote: "One of our concerns about non-backtracking counterfactuals, despite their novelty and elegance, is precisely their frequent requirement to do out-of-sample generalization." Out-of-sample generalization is not a concern of non-backtracking (interventional) counterfactuals; instead, it is a necessary feature of it.**
>
> As explained in Question W(2)-1 above, being out-of-sample (in the sense used in our paper) is NOT a necessary feature of non-backtracking counterfactuals. For example, unlike $do(n_2 = 0.19)$ given the evidence $(n_1, n_2, n_3) = (-0.59, 0.71, -0.37)$, the intervention $do(n_2 = 0.50)$ is not out-of-sample, meaning that $(n_1 = -0.59, n_2 = 0.50)$ falls within the support of the observable distribution.
>
> Additionally, out-of-sample scenarios can pose challenges under certain conditions. For example, non-parametric models routinely struggle and often fail to generalize to out-of-sample data points, as demonstrated in our experiments, among others.
>
> **Q(3): I read [30] again. They have Property 1 (preference for closeness). In other words, the closest (most similar) world to an actual world is itself. Please note that this is not important for the main issues of the paper.**
>
> Yes, property 1 in [30] states that the highest probability density is assigned to the same value. However, during inference, they will sample not only the data point with the highest probability but also other data points with lower probability density. Technically, since [30] assumes continuous variables, the probability of sampling the exact closest world is zero (e.g., the value $a$ when $a^* = a$), whereas the probability is positive for sampling values around $a$. But we agree this point is not important for their main purposes.
>
>
> ``Thank you again for your useful feedback and precious time! We are eager to have further discussions with you and address your remaining concerns, if any.``

---

> > ### Comment · Reviewer_JBox · 2024-08-10
> >
> > Thank you for your elaborate response! I do not have any further questions at this point.

---

> ### Author Response · Authors · 2024-08-10
> **Thank you again and please consider updating the score**
>
> Dear Reviewer JBox,
>
> Thank you once again for your prompt feedback. We hope your concerns have been addressed. Your comments have substantially helped improve our paper.
>
> If you think your questions have been properly addressed, we would be immensely grateful if you could *reconsider and update your recommendation*.
>
> If there are other questions, we hope to have opportunities to respond to them.
>
>
> Sincerely,
>
> Authors

---

> > ### Comment · Reviewer_JBox · 2024-08-12
> >
> > Based on your technical efforts, I’ve decided to increase my rating. However, in my humble opinion, the following two main concerns remain:
> >
> > (1) The terms "non-backtracking" and "backtracking" have specific meanings in fields like ML, philosophy, and cognitive science. To avoid confusion, it might be better to talk only about a new concept "Natural Counterfactuals." For instance, backtracking refers to observational counterfactuals and deals with use cases like, "Had I observed $n_2=0.19$, what would $n_3$
> > be?". Here, you track the difference between factual and counterfactual values back to the parents. In contrast, your paper begins by requesting a change to $n_2=0.19$, and then your optimization framework finds new values for both $n_1$ and $n_2$, resulting in a simultaneous change for two variables. This approach doesn't align with the established definitions of backtracking or non-backtracking.
> >
> > (2) What specific problem or use case does your approach address? For example, in lines 50-51, you mention autonomous driving. If the observational distribution is sunny weather with 40°C, then asking what would happen if the weather changed to heavy rain is a valid non-backtracking counterfactual, which would likely involve out-of-distribution samples. Replacing it with cloudy weather and 35°C, which is closer to the observational distribution, doesn't fully address the original question.
> >
> > The XAI recourse literature uses optimization methods to find counterfactual values based on causal structures. While these techniques are typically applied in supervised learning, similar use cases may exist in other areas, such as generative AI (GenAI).

---

> > > ### Author Response · Authors · 2024-08-13
> > > **Thank You for Your Kind Feedback**
> > >
> > > Thank you very much for increasing your rating, and for sharing your remaining concerns.
> > >
> > > Regarding Question/Suggestion (1), we will follow your advice to focus more on the concept of "natural counterfactuals" and be clearer about the sense of backtracking used in this paper. With due respect, we do not think the term "backtracking" has become so fixed to a specific meaning in a specific approach (among many possible alternatives) that its general meaning of "not keeping the temporal or causal upstream fixed in making counterfactual supposition" should not be used anymore. But thank you for your reminder of the potential of causing confusion; we will make extra efforts to make the meanings of terms clear.
> > >
> > > For Question (2), we agree that our approach does not fully address the original question in that example, if the original question is a non-backtracking interpretation of the counterfactual query and the non-backtracking interpretation turns out to violate our standard for picking out an intervention to realize the counterfactual supposition. That is NOT our aim. Our motivation is that in such cases in which we have good reasons to doubt that we can answer the non-backtracking interpretation of a counterfactual query (say, what if it were raining hard *and, implicitly, other variables in its causal upstream were kept the same*?) reliably based on available data, instead of returning a bad answer or simply saying "do not know", we can also (or even should) tell the user that if they adopt an answerable interpretation of the question that involves some backtracking (say, what if it were raining hard *and the atmospheric pressure were low*? We use atmospheric pressure instead of temperature in our example because atmospheric pressure is more likely to be a causal ancestor of weather in the given causal diagram), there is an answer we can reliably offer and here is the answer. Moreover, the user may have started with such an interpretation to begin with, as counterfactual queries are often ambiguous or under-specified. In any case, we reiterate that we are dealing with a distinctive interpretation of a counterfactual query and do not aim to address the "original" non-backtracking interpretation of the query, when that "original question" is demonstrably hard to answer given the available data.
> > >
> > > Thank you once again for your time and excellent questions!

---

> ### Author Response · Authors · 2024-08-12
> **Your Feedback Would Be Appreciated**
>
> Dear Reviewer JBox,
>
> Thank you once again for your valuable comments. Your suggestions on clarifying our motivations and inference procedure were very helpful. We are eager to know if our responses have adequately addressed your concerns.
>
> Due to the limited time for discussion, we look forward to receiving your feedback and hope for the opportunity to respond to any further questions you may have.
>
> Yours Sincerely,
>
> Authors of Submission 4096

---

### Official Review · Reviewer_v2As · 2024-07-12

**Soundness:** 2
**Presentation:** 3
**Contribution:** 3
**Rating:** 6
**Confidence:** 4

**Summary:**

The paper takes the recently developed idea of backtracking counterfactuals and applies it to improve the realistic generation of counterfactuals from data, which is known to be a hard task as the standard, non-backtracking, counterfactuals lie out of the distribution and generative models perform badly on those.

**Strengths:**

The idea of using backtracking counterfactuals as being more "natural", and then invoking them to improve the generation of counterfactuals when the full SCM is not available, is a very good one.

**Weaknesses:**

There are some technical issues which need to be addressed before the paper can be accepted.

**Questions:**

Full disclosure: I reviewed a previous version of this paper for last year's conference, and the current version is much much better. Still, there are some issues which the authors should address. Some of them might be mistakes on my part due to my misunderstanding, others are questions for clarification, others are problems that need to be solved.

First I present some larger issues, followed by some minor issues as they appear in the order of the paper.

1: I find the motivation rather odd, which is that the supposed benefit of natural counterfactuals is that they are easier to learn. But surely the main focus should be: which counterfactual (backtracking, non-backtracking, partial backtracking) is appropriate/meaningful in a given situation, rather than “which one can we learn”. Simply put, we shouldn't just focus on learning something because it's easy to learn, the thing learned should also be sensible. The paper actually remains silent on this issue, or rather, it seems to imply that non-backtracking counterfactuals are always the only sensible ones, and their natural counterfactuals are therefore just a heuristic. The reason I say this is implied is because in the experiments the non-backtracking one is taken as the ground truth. This is at odds with the related literature cited, which in fact argues that sometimes the backtracking counterfactual is the "ground truth", i.e., semantically it is what is meant with a counterfactual query. It would be good if the authors could be more explicit on where they stand on this issue.

2: There are several implicit assumptions that should be addressed.

- It is assumed that there is a unique an(A)*, yet the definitions given do not guarantee unicity.

- Initially it seems as if it is also assumed that an an(A)* always exists, which is not the case. Although this is made clear later, it would be good to flag this early on.

3: 159: This I don’t understand, because it violates the idea that C has to include A. Take the trivial case in which a*=a.  Then you will get the empty LBF intervention, which of course does not include A.

4: Building on the previous point, what if we have A=a, and yet A=a was extremely unlikely (so in the far-end of the tail of the distribution). Now consider the counterfactual a*=a. Will we not get that there is no solution? And if so, isn't that strange? This could be mitigated by combining the distance criterion and the naturalness criterion in a more weighted fashion, rather than having the latter be a necessary criterion and only then apply the distance one. This would bring it more in line with [30], see their 3.17 in particular.

5: Continuing, note that Choice (2) seems to result in a variation of 3.17, except that the distance is here between endogenous variables. Note though that a distance on endogenous variables can easily induce a distance between exogenous variables, by considering the two endogenous states that would result from two exogenous states, and thus this is not an essential difference.

6: Th4.1 is described as being about the identifiability of counterfactuals, but it is not, because it already assumes knowledge of do(C=c*), and that is part of what needs to be identified. Furthermore, it is then said that the theorem confirms that counterfactuals fall within the support of the observational distribution, but this ignores the earlier point that an LBF need not exist, and thus we are only guaranteed to identify counterfactuals in the case that it does.

7: I did not have time to closely examine the experiments, except for the following: all the results are only about those counterfactuals which happen to be in the scope of the natural counterfactuals, and the others are excluded. Yet it seems crucial to know how many were excluded in this manner in order to evaluate how useful in practice this method is, so this should be reported as well.

Minor issues:

81: "a most recent paper" I’d say a more accurate description is that [30] was the first paper to formally introduce backtracking counterfactuals within the causal models framework, and thus the current paper builds on that one. (Also, the usefulness for counterfactual explanations is also something that is explicitly discussed in [30].)

100: So the paper is limited to assuming independent and unique exogenous variables. Note that this is not the case in [30].

121: "In this paper..." Why? The generalization seems trivial.

175: This assumes that all variables are real-valued. Yet later it is mentioned that not all variables need to be at the same scale. Isn't that inconsistent?

221: Why restrict to a single variable A all of a sudden?

Th4.1: Doesn't the independence of U_i and Pa_i already follow from the independence of the exogenous variables?

Th4.1: Why is there no mention of the distance criterion for do(C=c*)?


Typos:

110: "distribution" -> "distributions"

123: "date"

234: "encourage"

303: "datasets, which"

**Limitations:**

Yes

---

> ### Author Rebuttal · Authors · 2024-08-07
>
> We are very grateful for your previous feedback, which helped a lot to improve the paper. Thank you also for the new comments. We will address them and correct the typos you pointed out.
>
> **1: Surely the main focus should be: which counterfactual (backtracking, non-backtracking, partial backtracking) is appropriate/meaningful in a given situation, rather than “which one can we learn”. The reason I say this is implied is because in the experiments the non-backtracking one is taken as the ground truth.**
>
> Thank you for this insightful comment, which has helped make our motivation more clearly articulated. It is a consequence of our motivation that our counterfactuals are more learnable than non-backtracking counterfactuals, but being easier to learn is not the primary motivation. We are mainly motivated by observing that thoroughly non-backtracking counterfactual generation, for all its merits, often results in scenarios that are too unrealistic (and so, as a consequence, unreliable when it is based on training data). On the other hand, however, we also think that fully backtracking counterfactuals may not have much implication for guiding actions or decision-making, because they appeal to inventions in unobserved or even unobservable, unidentifable variables. We thus aim to develop a framework that generates sufficiently realistic counterfactuals by finding feasible interventions on observed variables while minimizing the extent of backtracking. In particular, when the intervention set $C$ picked out by our algorithm is sufficiently simple (a special case is that it is identical to the original target set $A$), the counterfactual will remain useful for guiding actions.
>
> *In addition, we hasten to clarify that we did not use non-backtracking counterfactuals as the ground truth.* We apply the same trained models for inferring non-backtracking and natural counterfactuals. In toy experiments, a ground-truth SCM serves as the gold standard to measure the error of inference. Specifically, we compare the ground-truth outcomes with the predicted outcomes after intervening on $A$ in non-backtracking counterfactuals or intervening on $C$ in our natural counterfactuals. In other words, non-backtracking counterfactuals and natural counterfactuals have different ground truths, and the experiments illustrate that the generative algorithm for the former deviates more from its ground truth than our generative algorithm does from the latter's ground truth. In real-world datasets, due to the lack of a ground-truth SCM, we measure how accurately predicted outcomes reflect desired image attributes, which are represented by the input values stated in the antecedents of the counterfactuals, for example, $t = 3$, meaning that thickness should be $3$.
>
> **2: It is assumed that there is a unique $an(A)^\*$, yet the definitions given do not guarantee unicity.**
>
> Thank you for pointing this out. We do not assume that $an(A)^*$ is unique. In fact, $an(A)^*$ can theoretically have multiple solutions (or no solution at all as mentioned in Section 5). We have made this clearer in our updated paper. Even though $an(A)^*$ can have multiple solutions, the inference method remains the same after we sample one value from the available solutions.
>
> **3: 159: This I don’t understand, because it violates the idea that C has to include A. Take the trivial case in which a\*=a. Then you will get the empty LBF intervention, which of course does not include A.**
>
> We follow the standard treatment by requiring that even when the counterfactual value $a^*$ is equal to the actual value $a$, an intervention would still be invoked to realize $a^*=a$ (among other things, the links between $A$ and its parents would still be severed). Hence, in the special case when $a^*=a$, $C$ still contains $A$.
>
> **4: Building on the previous point, what if we have A=a, and yet A=a was extremely unlikely (so in the far-end of the tail of the distribution). Now consider the counterfactual a\*=a. Will we not get that there is no solution?**
>
> Excellent point! In natural counterfactuals, we use a hyperparameter $\epsilon$ to determine whether a point is considered natural. Therefore, in this situation, there will be no solution if $A = a$ is so unlikely that it conflicts with $\epsilon$-natural generation. This is a consequence of introducing a parameter to control the degree of naturalness. However, if we only require the lowest level of naturalness, where $\epsilon$ is zero (i.e., we only require that counterfactuals are within the distribution support), this situation will be eliminated. Otherwise, yes, there could turn out to be no solutions for natural counterfactuals. As you noticed, this may happen in extreme situations, which are not "natural" themselves and already deserve extra inspection and attention.  We have made it clearer in the paper.
>
>
> **5: Continuing, note that Choice (2) seems to result in a variation of 3.17, except that the distance is here between endogenous variables.**
>
> Thank you for this interesting comparison. Here we do not consider Choice (2) to be a variation of 3.17, because while 3.17 is based on value distance, Choice (2) relies on CDF distance. This distinction reflects a key shortcoming of 3.17: the distribution of exogenous variables is not theoretically identifiable, making it challenging to apply in practice. However, due to the properties of CDF distance, Choice (2) allows us to avoid calculating distances related to exogenous variables. Thus, Choice (2) can be computed without considering the distributions of exogenous variables.
>
> ***``Due to space limitations, we have included questions 6 and 7, along with minor issues, in the global response.``***

---

> > ### Comment · Reviewer_v2As · 2024-08-11
> > **Reply to rebuttal**
> >
> > I thank the authors for their rebuttal. It has prompted some more questions.
> >
> > Re1:
> > Thanks for the clarification. It still does not really address the question what the correct semantics for counterfactuals is though.
> >
> > 1.1: You speak of the non-backtracking counterfactual being "unrealistic", but here unrealistic seems to mean: if we try to estimate the standard, Pearl-style, interventional counterfactual without access to the ground truth SCM, we get bad answers. So the underlying semantics is still the standard interventional one, it's just that we don't have a good estimator for it. (If I understand correctly, the ground truths for both the non-backtracking and the natural counterfactuals are both expressed in terms of an interventional counterfactual, it's just that they are compared to different interventions.)
> >
> > 1.2: "because they appeal to interventions..." Fully backtracking counterfactuals do not appeal to interventions at all, on the contrary, they're directly at odds with the interventional semantics of Pearl. Instead, they appeal to a change in the initial conditions, which is not an intervention because the initial conditions are not determined by causal equations and thus no "laws of nature" are broken. So aside from the issue as to which styles of counterfactuals are easier to estimate without access to the ground truth SCM, there is the more fundamental issue of which type of counterfactual is the formal representation of the informal scientific or even natural language query that we are formalizing.
> >
> > 1.3: To be clear, the above is not a criticism of the method itself, I'm simply trying to get conceptual clarity on what the assumed true underlying semantics for counterfactuals is. I guess this is the question that would resolve it: say you _do_ have access to the ground truth SCM, and thus perfect inferences can be made. And now we want to reason about some counterfactual "If A were a*". How would you compute it? Using do(A=a*), or using change(A=a*)?
> >
> > Re: 3: You should change the wording, because it does not match your explanation. The difference between a and a is the emptyset.

---

> ### Author Response · Authors · 2024-08-12
> **Response 1**
>
> We are grateful for your time and insights, and really appreciate the opportunity to further clarify our responses to your questions.
>
> **1.1: You speak of the non-backtracking counterfactual being "unrealistic", but here unrealistic seems to mean: if we try to estimate the standard, Pearl-style, interventional counterfactual without access to the ground truth SCM, we get bad answers. So the underlying semantics is still the standard interventional one, it's just that we don't have a good estimator for it. (If I understand correctly, the ground truths for both the non-backtracking and the natural counterfactuals are both expressed in terms of an interventional counterfactual, it's just that they are compared to different interventions.)**
>
> This is an excellent point. Our semantics is indeed still based on interventions, and you are exactly right that the difference between ours and the standard Pearlian semantics is that they (often) pick out different interventions to realize the counterfactual supposition in the antecedent of a counterfactual conditional. In other words, given a counterfactual query: what if $A = a^*$?, we have different interpretations on how this change is supposed to be realized (or how the supposition $A=a^*$ is to be further specified). In Pearl's semantics, the interpretation is simple and elegant: The change is to be realized by an intervention on A alone (or the supposition is to be specified as "$A=a^*$ *and every variable in the causal upstream of A remains the same*"); in our semantics, by contrast, we impose a standard for picking out an intervention to realize the supposition, which may require intervening on some of $A$'s causal antecedents as well (and in this sense involve backtracking). So, although they are both based on interventions, they still constitute different interpretations of the original counterfactual supposition.
>
> Regarding our claim that the non-backtracking counterfactuals are sometimes "unrealistic", you gave a very nice statement of one important reason why we think this. For the purpose of this paper, perhaps this epistemic motivation is the most relevant and easiest to see, and we will consider focusing on this motivation and elaborating on its significance. In your original review, you seemed to indicate that this motivation is not sufficiently compelling, on which we respectfully disagree. If reliable answers to a counterfactual query interpreted in a certain way are very difficult or even impossible to obtain, it seems to us an excellent reason to develop a different but related interpretation that can still serve the desired functions to a good extent and admit more reliable answers. Imagine someone comes with a question: If his weight were decreased to 70kg, what would be his risk of diabetes? Suppose, given the causal assumptions and data we have, we cannot reliably answer the non-backtracking interpretation of this question. It seems that instead of just saying that we do not know, we can or even should adopt an answerable interpretation of the question and tell him something we can reliably offer, say, if he decreased weight to 70kg *and exercised for more than 150 minutes a week*, his risk would be such and such.
>
> For what it is worth, we also think a perhaps more important reason for sometimes regarding non-backtracking counterfactuals as "unrealistic" is that their way of interpreting the counterfactual supposition, say, $A=a^*$, as intervening on A alone while keeping every causal ancestor in the model intact, may be extremely difficult or practically impossible to realize (e.g., having a person stand still on a bus while keeping the sudden braking in the toy example we mentioned in the paper), as indicated by the rarity of the scenario in the data, and for that reason is not very relevant or helpful for guiding actual actions or decision-making. Imposing a naturalness criterion is one way to ensure that the picked out interventions are at least feasible to carry out in light of the evidence we have.

---

> ### Author Response · Authors · 2024-08-12
> **Response 2**
>
> **1.2: "because they appeal to interventions..." Fully backtracking counterfactuals do not appeal to interventions at all, on the contrary, they're directly at odds with the interventional semantics of Pearl. Instead, they appeal to a change in the initial conditions, which is not an intervention because the initial conditions are not determined by causal equations and thus no "laws of nature" are broken. So aside from the issue as to which styles of counterfactuals are easier to estimate without access to the ground truth SCM, there is the more fundamental issue of which type of counterfactual is the formal representation of the informal scientific or even natural language query that we are formalizing.**
>
> We agree. In our view, informal scientific and natural language queries in the form of counterfactual conditionals are very often ambiguous or under-specified (in their antecedents), the disambiguation or specification of which depends on contexts. We do not pretend to be able to resolve this fundamental issue. We also agree that fully backtracking counterfactuals do not appeal to interventions in the usual Pearlian sense, for those are restricted to endogenous variables and are associated with breaking endogenous mechanisms. However, formally speaking, changing the values of exogenous variables is analogous to intervening on exogenous variables, in the sense that they change those variables without affecting the mechanisms for other variables in the model (the invariance of the mechanisms for other variables is for many the crucial hallmark of an intervention). In fact, some authors explicitly advocate applying the notion of intervention to exogenous variables as well. It is in this spirit that we say fully backtracking counterfactuals appeal to "interventions" to exogenous variables. We now see that it is potentially misleading, and we will be more careful with the wording.
>
>
> **1.3: To be clear, the above is not a criticism of the method itself, I'm simply trying to get conceptual clarity on what the assumed true underlying semantics for counterfactuals is. I guess this is the question that would resolve it: say you do have access to the ground truth SCM, and thus perfect inferences can be made. And now we want to reason about some counterfactual "If A were a\*". How would you compute it? Using do(A=a\*), or using change(A=a\*)?**
>
> In such a case, yes, it will depend on how to interpret the counterfactual supposition, and we agree that in many contexts, the non-backtracking interpretation will probably stand out as the most salient (or as the philosopher David Lewis once wrote, "standard") resolution. In other words, in such a case, the epistemic motivation for adopting our interpretation is of course annulled. Still, the more ontic motivation described above, having to do with the feasibility of carrying out a non-backtracking intervention, may still be relevant.
>
> Thank you again for making these important and subtle questions so clear. We will improve our statements of the motivations accordingly.
>
>
> **Re: 3: You should change the wording, because it does not match your explanation. The difference between a and a is the emptyset.**
>
> That's right. It is our sloppiness and thank you for catching it. We will rephrase carefully.
>
>
> ``We look forward to receiving your feedback and would appreciate the opportunity to answer any additional questions you may have.``

---

> > ### Comment · Reviewer_v2As · 2024-08-12
> >
> > Thanks for these further clarifications.
> >
> > I still have some reservations about your description of fully backtracking counterfactuals, because their philosophical motivation was precisely to avoid anything even resembling an intervention, (strictly speaking, there is backtracking all the way to the initial conditions of the universe, it just so happens that in causal models the buck stops at the exogenous variables), but I think this is an issue that is mostly orthogonal to the paper and thus not that relevant.
> >
> > I agree that the epistemic perspective does in and of itself justify the value of your contribution, and I would even suggest that in the final version you try to remain as neutral as possible regarding what the correct semantics for counterfactuals is (or even what the correct semantics is in some specific context), because the contribution of the paper is not really about that and therefore the less you assume the better. For what it's worth, personally I do expect that in some contexts, your partially backtracking counterfactuals are also the correct counterfactuals, in the sense that people would indeed be inclined to interpret counterfactuals in that manner if the circumstances warrant. But perhaps that's something to evaluate in future work, together with some psychologists.

---

> > > ### Author Response · Authors · 2024-08-12
> > > **Your Feedback Is Much Appreciated**
> > >
> > > Thank you so much for all of your feedback and for appreciating the value of this work. We greatly enjoyed our discussions and learned a lot.
> > >
> > > We will avoid the misleading characterization of fully backtracking counterfactuals as appealing to interventions on exogenous variables, and will make it clear that our aim is not to propose a "correct" semantics for counterfactuals, but instead one that is useful in some situations and for some purposes.
> > >
> > > Once again, we are very grateful for your time and insights.

---

### Official Review · Reviewer_DQof · 2024-07-13

**Soundness:** 3
**Presentation:** 3
**Contribution:** 3
**Rating:** 6
**Confidence:** 3

**Summary:**

This paper addresses a key limitation of non-backtracking counterfactual reasoning in causal inference. The authors argue that Pearl's framework often generates unrealistic scenarios. To solve this, they propose "natural counterfactuals," which allow controlled backtracking to ensure scenarios remain realistic.They introduce Feasible Intervention Optimization (FIO), a novel framework that generates natural counterfactuals by incorporating a naturalness constraint. This ensures counterfactual instances are plausible given the observed data distribution. The authors also propose distance measures to minimize backtracking while achieving the desired outcome.

**Strengths:**

- The paper is well-written.

- The paper identifies a crucial issue with non-backtracking counterfactuals, highlighting the importance of generating realistic scenarios.

- FIO provides a principled and practical method for generating natural counterfactuals with clear mathematical structure.

- The paper presents convincing experiments on both simulated and real-world datasets, demonstrating the effectiveness of the proposed approach.

**Weaknesses:**

- FIO involves several parameters and choices (naturalness criteria, distance measures), which may require careful tuning and consideration for different applications.

- The paper mainly focuses on contrasting natural counterfactuals with non-backtracking ones. A more thorough comparison with existing backtracking approaches is missing in the experiment section.

- The paper mainly focuses on contrasting natural counterfactuals with non-backtracking ones. A more thorough comparison with existing backtracking approaches is missing in the experiment section.

**Questions:**

- How sensitive is the proposed method to violations of the assumptions about the underlying SCM?

- What is the computational complexity of FIO and how does it scale with the size of the causal graph and the dataset?

**Limitations:**

See weaknesses and questions

---

> ### Author Rebuttal · Authors · 2024-08-07
>
> ***Weaknesses***
>
> **1. FIO involves several parameters and choices (naturalness criteria, distance measures), which may require careful tuning and consideration for different applications.**
>
> Thanks for raising this point. We agree that different applications may warrant different choices, as stated in our Conclusion section. Our main purpose is to provide a general and flexible framework for generating natural or realistic counterfactuals with minimal backtracking. The exact choices of parameters and measures can and should be tailored to specific applications, as you suggested.
>
> **2. The paper mainly focuses on contrasting natural counterfactuals with non-backtracking ones. A more thorough comparison with existing backtracking approaches is missing in the experiment section.**
>
> Thank you for the suggestion. In light of them, we have added such comparisons to our updated paper. Backtracking Counterfactuals in [30] assume a known joint distribution over factual and counterfactual exogenous variables, which is often impractical due to the non-identifiability of the exogenous distribution. In contrast, we only require access to the observed distribution. In our simulation experiments, for the purpose of implementing Backtracking Counterfactuals, we assume the joint distribution over factual and counterfactual exogenous variables is known. the results show that backtracking counterfactuals can generate infinite counterfactuals for a single query, with higher probability given to those more similar to the actual instance. Conversely, our approach typically produces a single counterfactual closest to the actual instance. Note that an extreme case illustrates that [30] sometimes invokes gratuitous changes. When the counterfactual value $a^*$ equals the actual value $a$, the counterfactual instance should logically match the actual one. However, [30] also assigns positive probability to other instances.
>
> Reference:
>
> [30] Julius von Kügelgen, Abdirisak Mohamed, and Sander Beckers. Backtracking counterfactuals. arXiv preprint arXiv:2211.00472, 2022.
>
> ***Questions***
>
> **1. How sensitive is the proposed method to violations of the assumptions about the underlying SCM?**
>
> As widely seen in the literature of causal inference and counterfactual inference, certain assumptions are essential. For Theorem 4, if the assumptions do not hold, identifiability may not be guaranteed. For example, suppose $Y=XU_1+U_2$ where $Y$ and $X$ are endogenous variables and $U_1$ and $U_2$ are exogenous noises, then the counterfactual outcome will not be identifiable.
>
> **2. What is the computational complexity of FIO and how does it scale with the size of the causal graph and the dataset?**
>
> The computational complexity of FIO scales linearly with both the size of the causal graph (specifically, the number of ancestors of $A$) and the dataset size. Formally, the overall complexity is $O(KPTM)$, where $K$ is the number of ancestors of $A$, $P$ is the number of parameters in the neural networks, $T$ is the number of optimization steps, and $M$ is the number of data points in the dataset. This linear scaling indicates that FIO is reasonably scalable, making it suitable for large causal graphs and datasets.

---

### Author Rebuttal · Authors · 2024-08-07

We sincerely thank all the reviewers for their time dedicated to reviewing this paper and their valuable comments. We are encouraged by and grateful for the comments that say our paper was described as  ``"well-written"``  (DQof) and ``"interesting"`` (fYWn), and was considered to``"make a strong contribution"`` (fYWn). Additionally, we appreciate the recognition of our approach as ``"a principled and practical method"`` with a ``"clear mathematical structure"`` (DQof), and as ``"a very good one"`` (v2As) and ``"interesting"`` (fYWn).
The acknowledgment of our empirical results demonstrating ``"convincing experiments"`` (DQof) and ``"improved performance"`` (fYWn) is particularly exciting.

We have carefully responded to all important questions and hope to properly address any remaining concerns.

---------------

***Below are additional responses to questions 6 and 7, as well as minor issues, for Reviewer v2As.***

**6: Th4.1 is described as being about the identifiability of counterfactuals, but it is not, because it already assumes knowledge of do(C=c\*), and that is part of what needs to be identified. Furthermore, this ignores the earlier point that an LBF need not exist, and thus we are only guaranteed to identify counterfactuals in the case that it does.**

Thank you for this valuable comment. When discussing the identifiability of natural counterfactuals, we assume that a natural counterfactual exists, meaning that $C$ has at least one solution. Therefore, we do not address the distance criterion for $do(C=c^*)$ and the solutions of $C$ directly. Theoretically, $C$ may have one or more identifiable solutions, or no solution at all, and this can always be identified. For multiple solutions, we sample one value from these solutions, and its counterfactual is identifiable according to Theorem 4.1.

We have clarified this in our updated manuscript.

**7: I did not have time to closely examine the experiments, except for the following: all the results are only about those counterfactuals which happen to be in the scope of the natural counterfactuals, and the others are excluded. Yet it seems crucial to know how many were excluded in this manner in order to evaluate how useful in practice this method is, so this should be reported as well.**

We present this result in Table 6 on Page 14, where we report the frequency of unfeasible solutions per 10,000 instances in the MorphoMNIST dataset. The data reveals a consistent trend: as the value of $\epsilon$ increases, the frequency of unfeasible solutions also rises. This occurs because a higher $\epsilon$ corresponds to a stricter standard of naturalness. In theory, when $\epsilon = 0$ (i.e., we only require that counterfactuals are within the distribution support), there are no unfeasible solutions.

| $\epsilon$ | Unfeasible Solutions |
|:------------:|:---------------------:|
|     1e-4     |          794          |
|     1e-3     |          975          |
|     1e-2     |         1166          |



***Minor issues***

**81: "a most recent paper" I’d say a more accurate description is that [30] was the first paper to formally introduce backtracking counterfactuals within the causal models framework, and thus the current paper builds on that one. (Also, the usefulness for counterfactual explanations is also something that is explicitly discussed in [30].)**

Thanks for this suggestion. We will add an accurate description of [30], as the first paper to formally introduce (fully) backtracking counterfactuals within the SCM framework. At the same time, we wish to add that our work differs significantly in both the motivation and the approach. We aim to find feasible interventions to make counterfactuals more realistic by intervening on endogenous variables. In contrast, [30] assumes a known exogenous distribution and always intervenes on exogenous variables. We explain the key differences between our work and [30] in Section E2 of the Appendix.

**100: So the paper is limited to assuming independent and unique exogenous variables. Note that this is not the case in [30].**

We completely agree. One purpose of using the independence assumption is to ensure the identifiability of natural counterfactuals when we only have access to endogenous variables. If we also assume a known distribution of exogenous variables, our work can be generalized.


**121: "In this paper..." Why? The generalization seems trivial.** and **221: Why restrict to a single variable A all of a sudden?**

We agree. $A$ could be a set of variables. In fact, in our experiment on the 3DIdentBox dataset, we perform interventions on a set of three variables. In the paper, for the sake of simplicity, we initially assume $A$ is a single variable. We have modified our presentation to explicitly state that $A$ can be a single variable or a set of variables.

**175: This assumes that all variables are real-valued. Yet later it is mentioned that not all variables need to be at the same scale. Isn't that inconsistent?**


We meant to say that all variables are continuous, and scale refers to the range of support. For example, $V_1$ follows a normal distribution with a range of $(-\infty, \infty)$. $V_2$ follows a uniform distribution over the interval $[0, 1]$. We have clarified this point in our updated manuscript.


**Th4.1: Doesn't the independence of U_i and Pa_i already follow from the independence of the exogenous variables?**

That is true. We have updated the presentation accordingly. Thank you.

**Th4.1: Why is there no mention of the distance criterion for do(C=c\*)?**

Please see our response to Q6 above. We have clarified it in the updated manuscript.

---

### Decision · Program_Chairs · 2024-09-25

**Decision:**

Accept (poster)

**Comment:**

The reviewers and authors held a vigorous discussion. Overall the consensus is that the paper presents a novel approach to an important problem. The approach is especially promising in the era of big generative model, as the problem of generating "natural counterfactuals" comes up in many applications of such models. During the discussion multiple comments were made that would improve the paper, including its motivation and presentation - I trust the authors will follow through on these in the finalized version.